# Robust Distributed Learning: Tight Error Bounds and Breakdown Point under Data Heterogeneity

**Youssef Allouah**[*]    **Rachid Guerraoui**    **Nirupam Gupta**    **Rafaël Pinot**    **Geovani Rizk**

Ecole Polytechnique Fédérale de Lausanne (EPFL), Switzerland

## Abstract

The theory underlying robust distributed learning algorithms, designed to resist adversarial machines, matches empirical observations when data is *homogeneous*. Under *data heterogeneity* however, which is the norm in practical scenarios, established lower bounds on the learning error are essentially vacuous and greatly mismatch empirical observations. This is because the heterogeneity model considered is too restrictive and does not cover basic learning tasks such as least-squares regression. We consider in this paper a more realistic heterogeneity model, namely $(G, B)$-gradient dissimilarity, and show that it covers a larger class of learning problems than existing theory. Notably, we show that the breakdown point under heterogeneity is lower than the classical fraction $1/2$. We also prove a new lower bound on the learning error of any distributed learning algorithm. We derive a matching upper bound for a robust variant of distributed gradient descent, and empirically show that our analysis reduces the gap between theory and practice.

## 1   Introduction

Distributed machine learning algorithms involve multiple machines (or *workers*) collaborating with the help of a server to learn a common model over their collective datasets. These algorithms enable training large and complex machine learning models, by distributing the computational burden among several workers. They are also appealing as they allow workers to retain control over their local training data. Conventional distributed machine learning algorithms are known to be vulnerable to adversarial workers, which may behave unpredictably. Such behavior may result from software and hardware bugs, data poisoning, or malicious players controlling part of the network. In the parlance of distributed computing, such adversarial workers are referred to as *Byzantine* [24]. Due to the growing influence of distributed machine learning in public applications, a significant amount of work has been devoted to addressing the problem of robustness to Byzantine workers, e.g., see [34, 11, 2, 19, 13, 15].

A vast majority of prior work on robustness however assumes *data homogeneity*, i.e., local datasets are generated from the same distribution. This questions their applicability in realistic distributed learning scenarios with *heterogeneous data*, where local datasets are generated from different distributions. Under data homogeneity, Byzantine workers can only harm the system when the other workers compute stochastic gradient estimates, by exploiting the noise in gradient computations. This vulnerability can be circumvented using variance-reduction schemes [19, 13, 14]. In contrast, under data heterogeneity, variance-reduction schemes are not very helpful, as suggested by preliminary work [12, 20, 3]. In short, data heterogeneity is still poorly understood in robust distributed learning. In particular, existing robustness guarantees are extremely conservative, and often refuted by empirical observations. Indeed, the heterogeneity model generally assumed is typically violated in practice and does not even cover basic machine learning tasks such as least-squares regression.

---

[*]Correspondence to: Youssef Allouah <youssef.allouah@epfl.ch>.

37th Conference on Neural Information Processing Systems (NeurIPS 2023).

Our work addresses the aforementioned shortcomings of existing theory, by considering a more realistic heterogeneity model, called $(G, B)$-*gradient dissimilarity* [18]. This criterion characterizes data heterogeneity for a larger class of machine learning problems compared to prior works [12, 20, 3], and enables us to reduce the gap between theoretical guarantees and empirical observations. Before summarizing our contributions in Section 1.2, we briefly recall below the essentials of robust distributed learning and highlight the challenges of data heterogeneity.

## 1.1  Robust distributed learning under heterogeneity

Consider a system comprising $n$ workers $w_1, \ldots, w_n$ and a central server, where $f$ workers of a priori unknown identity may be Byzantine. Each worker $w_i$ holds a dataset $\mathcal{D}_i$ composed of $m$ data points from an input space $\mathcal{X}$, i.e., $\mathcal{D}_i := \{x_1^{(i)}, \ldots, x_m^{(i)}\} \in \mathcal{X}^m$. Given a model parameterized by $\theta \in \mathbb{R}^d$, each data point $x$ incurs a loss $\ell(\theta; x)$ where $\ell : \mathbb{R}^d \times \mathcal{X} \to \mathbb{R}$. Thus, each worker $w_i$ has a *local empirical loss* function defined as $\mathcal{L}_i(\theta) := \frac{1}{m} \sum_{x \in \mathcal{D}_i} \ell(\theta; x)$. Ideally, when all the workers are assumed *honest* (i.e., non-Byzantine), the server can compute a model minimizing the global average loss function given by $\frac{1}{n} \sum_{i=1}^{n} \mathcal{L}_i(\theta)$, without requiring the workers to share their raw data points. However, this goal is rendered vacuous in the presence of Byzantine workers. A more reasonable goal for the server is to compute a model minimizing the *global honest loss* [16], i.e., the average loss of the honest workers. Specifically, denoting by $\mathcal{H} \subseteq [n]$ where $|\mathcal{H}| = n - f$, the indices of honest workers, the goal in robust distributed learning is to solve the following optimization problem:[2]

$$\min_{\theta \in \mathbb{R}^d} \mathcal{L}_{\mathcal{H}}(\theta) := \frac{1}{|\mathcal{H}|} \sum_{i \in \mathcal{H}} \mathcal{L}_i(\theta) . \tag{1}$$

Because Byzantine workers may send bogus information and are unknown to the server, solving (even approximately) the optimization problem (1) is known to be impossible in general [26, 20]. The key reason for this impossibility is precisely data heterogeneity. Indeed, we cannot obtain meaningful robustness guarantees unless data heterogeneity is bounded across honest workers.

**Modeling heterogeneity.** Prior work on robustness primarily focuses on a restrictive heterogeneity bound we call $G$-*gradient dissimilarity* [12, 20, 3]. Specifically, denoting $\| \cdot \|$ to be the Euclidean norm, the honest workers are said to satisfy $G$-gradient dissimilarity if for all $\theta \in \mathbb{R}^d$, we have

$$\frac{1}{|\mathcal{H}|} \sum_{i \in \mathcal{H}} \|\nabla \mathcal{L}_i(\theta) - \nabla \mathcal{L}_{\mathcal{H}}(\theta)\|^2 \leq G^2 . \tag{2}$$

However, the above uniform bound on the inter-worker variance of local gradients may not hold in common machine learning problems such as least-squares regression, as we discuss in Section 3. In our work, we consider the more general notion of $(G, B)$-*gradient dissimilarity*, which is a prominent data heterogeneity model in the classical (Byzantine-free) distributed machine learning literature (i.e., when $f = 0$) [18, 23, 27, 29]. Recent works have also adopted this definition in the context of Byzantine robust learning [20, 14], but did not provide tight analyses, as we discuss in Section 5. Formally, $(G, B)$-gradient dissimilarity is defined as follows.

**Assumption 1** ($(G, B)$-gradient dissimilarity)**.** The local loss functions of honest workers (represented by set $\mathcal{H}$) are said to satisfy $(G, B)$-*gradient dissimilarity* if, for all $\theta \in \mathbb{R}^d$, we have[3]

$$\frac{1}{|\mathcal{H}|} \sum_{i \in \mathcal{H}} \|\nabla \mathcal{L}_i(\theta) - \nabla \mathcal{L}_{\mathcal{H}}(\theta)\|^2 \leq G^2 + B^2 \|\nabla \mathcal{L}_{\mathcal{H}}(\theta)\|^2 .$$

Under $(G, B)$-gradient dissimilarity, the inter-worker variance of gradients need not be bounded, and can grow with the norm of the global loss function's gradient at a rate bounded by $B$. Furthermore, this notion also generalizes $G$-gradient dissimilarity, which corresponds to the special case of $B = 0$.

## 1.2  Our contributions

We provide the first tight analysis on robustness to Byzantine workers in distributed learning under a realistic data heterogeneity model, specifically $(G, B)$-gradient dissimilarity. Our key contributions are summarized as follows.

---

[2]We denote by $[n]$ the set $\{1, \ldots, n\}$.
[3]The dissimilarity inequality is equivalent to $\frac{1}{|\mathcal{H}|} \sum_{i \in \mathcal{H}} \|\nabla \mathcal{L}_i(\theta)\|^2 \leq G^2 + (1 + B^2) \|\nabla \mathcal{L}_{\mathcal{H}}(\theta)\|^2$.

**Breakdown point.** We establish a novel *breakdown point* for distributed learning under heterogeneity. Prior to our work, the upper bound on the breakdown point was simply $\frac{1}{2}$ [26], i.e., when half (or more) of the workers are Byzantine, no algorithm can provide meaningful guarantees for solving (1). We prove that, under $(G, B)$-gradient dissimilarity, the breakdown point is actually $\frac{1}{2+B^2}$. That is, the breakdown point of distributed learning is lower than $\frac{1}{2}$ under heterogeneity due to non-zero growth rate $B$ of gradient dissimilarity. We also confirm empirically that the breakdown point under heterogeneity can be much lower than $\frac{1}{2}$, which could not be explained prior to our work.

**Tight error bounds.** We show that, under the necessary condition $\frac{f}{n} < \frac{1}{2+B^2}$, any robust distributed learning algorithm must incur an optimization *error* in

$$\Omega\left(\frac{f}{n - (2 + B^2)\, f} \cdot G^2\right) \tag{3}$$

on the class of smooth strongly convex loss functions. We also show that the above lower bound is tight. Specifically, we prove a *matching* upper bound for the class of smooth non-convex loss functions, by analyzing a robust variant of distributed gradient descent.

**Proof techniques.** To prove our new breakdown point and lower bound, we construct an instance of quadratic loss functions parameterized by their scaling coefficients and minima. While the existing lower bound under $G$-gradient dissimilarity can easily be obtained by considering quadratic functions with different minima and identical scaling coefficients (see proof of Theorem III in [20]), this simple proof technique fails to capture the impact of non-zero growth rate $B$ in gradient dissimilarity. In fact, the main challenge we had to overcome is to devise a *coupling* between the parameters of the considered quadratic losses (scaling coefficients and minima) under the $(G, B)$-gradient dissimilarity constraint. Using this coupling, we show that when $\frac{f}{n} \geq \frac{1}{2+B^2}$, the distance between the minima of the quadratic losses can be made arbitrarily large by carefully choosing the scaling coefficients, hence yielding an arbitrarily large error. We similarly prove the lower bound (3) when $\frac{f}{n} < \frac{1}{2+B^2}$.

### 1.3 Paper outline

The remainder of this paper is organized as follows. Section 2 presents our formal robustness definition and recalls standard assumptions. Section 3 discusses some key limitations of previous works on heterogeneity under $G$-gradient dissimilarity. Section 4 presents the impossibility and lower bound results under $(G, B)$-gradient dissimilarity, along with a sketch of proof. Section 5 presents tight upper bounds obtained by analyzing robust distributed gradient descent under $(G, B)$-gradient dissimilarity. Full proofs are deferred to appendices A, B and C. Details on the setups of our experimental results are deferred to Appendix D.

## 2 Formal definitions

In this section, we state our formal definition of robustness and standard optimization assumptions. Recall that an algorithm is deemed robust to adversarial workers if it enables the server to approximate a minimum of the global honest loss, despite the presence of $f$ Byzantine workers whose identity is a priori unknown to the server. In Definition 1, we state the formal definition of robustness.

**Definition 1 ($(f, \varepsilon)$-resilience).** A distributed algorithm is said to be $(f, \varepsilon)$-*resilient* if it can output a parameter $\hat{\theta}$ such that

$$\mathcal{L}_{\mathcal{H}}(\hat{\theta}) - \mathcal{L}_{*, \mathcal{H}} \leq \varepsilon,$$

where $\mathcal{L}_{*, \mathcal{H}} \coloneqq \min_{\theta \in \mathbb{R}^d} \mathcal{L}_{\mathcal{H}}(\theta)$.

Accordingly, an $(f, \varepsilon)$-resilient distributed algorithm can output an $\varepsilon$-*approximate* minimizer of the global honest loss function, despite the presence of $f$ adversarial workers. Throughout the paper, we assume that $\frac{f}{n} < \frac{1}{2}$, as otherwise $(f, \varepsilon)$-resilience is in general impossible [26]. Note also that, for general smooth non-convex loss functions, we aim to find an approximate *stationary point* of the global honest loss instead of a minimizer, which is standard in non-convex optimization [8].

**Standard assumptions.** To derive our lower bounds, we consider the class of smooth strongly convex loss functions. We derive our upper bounds for smooth non-convex functions, and for functions satisfying the Polyak-Łojasiewicz (PL) inequality. This property relaxes strong convexity, i.e., strong

convexity implies PL, and covers learning problems which may be non-strongly convex such as least-squares regression [17]. We recall these properties in definitions 2 and 3 below.

**Definition 2** (*L*-smoothness). A function $\mathcal{L}\colon \mathbb{R}^d \to \mathbb{R}$ is *L*-smooth if, for all $\theta, \theta' \in \mathbb{R}^d$, we have

$$\|\nabla\mathcal{L}(\theta') - \nabla\mathcal{L}(\theta)\| \leq L \|\theta' - \theta\| \ .$$

This is equivalent [28] to, for all $\theta, \theta'$, having $|\mathcal{L}(\theta') - \mathcal{L}(\theta) - \langle\nabla\mathcal{L}(\theta),\, \theta' - \theta\rangle| \leq \frac{L}{2} \|\theta' - \theta\|^2$.

**Definition 3** ($\mu$-Polyak-Łojasiewicz (PL), strong convexity). A function $\mathcal{L}\colon \mathbb{R}^d \to \mathbb{R}$ is $\mu$-PL if, for all $\theta \in \mathbb{R}^d$, we have

$$2\mu\left(\mathcal{L}(\theta) - \mathcal{L}_*\right) \leq \|\nabla\mathcal{L}(\theta)\|^2 \,,$$

where $\mathcal{L}_* := \min_{\theta \in \mathbb{R}^d} \mathcal{L}(\theta)$. Function $\mathcal{L}$ is $\mu$-strongly convex if, for all $\theta, \theta' \in \mathbb{R}^d$, we have

$$\mathcal{L}(\theta') - \mathcal{L}(\theta) - \langle\nabla\mathcal{L}(\theta),\, \theta' - \theta\rangle \geq \frac{\mu}{2} \|\theta' - \theta\|^2 \,.$$

Note that a function satisfies *L*-smoothness and $\mu$-PL inequality simultaneously only if $\mu \leq L$. Lastly, although not needed for our results to hold, when the global loss function $\mathcal{L}_\mathcal{H}$ is $\mu$-PL, we will assume that it admits a unique minimizer, denoted $\theta_*$, for clarity.

## 3 Brittleness of previous approaches on heterogeneity

Under the *G*-gradient dissimilarity condition, presented in (2), prior work has established lower bounds [20] and matching upper bounds [3] for robust distributed learning. However, *G*-gradient dissimilarity is arguably unrealistic, since it requires a *uniform* bound $G^2$ on the variance of workers' gradients over the parameter space. As a matter of fact, *G*-gradient dissimilarity does not hold in general for simple learning tasks such as least-squares regression, as shown in Observation 1 below.

**Observation 1.** In general, *G*-gradient dissimilarity (2) does not hold in least-squares regression.

*Proof.* Consider the setting given by $\mathcal{X} = \mathbb{R}^2$, $n = 2, f = 0$, and $d = 1$. For any data point $(x_1, x_2) \in \mathcal{X}$, consider the squared error loss $\ell(\theta; (x_1, x_2)) = \frac{1}{2}(\theta \cdot x_1 - x_2)^2$. Let the local datasets be $\mathcal{D}_1 = \{(1, 0)\}, \mathcal{D}_2 = \{(0, 1)\}$. Note that for all $\theta \in \mathbb{R}$, we have $\nabla\mathcal{L}_1(\theta) = \theta$, and $\nabla\mathcal{L}_2(\theta) = 0$. This implies that $\frac{1}{|\mathcal{H}|}\sum_{i \in \mathcal{H}} \|\nabla\mathcal{L}_i(\theta) - \nabla\mathcal{L}_\mathcal{H}(\theta)\|^2 = \theta^2/4$, where $\mathcal{H} = \{1, 2\}$, which is unbounded over $\mathbb{R}$. Hence, the condition of *G*-gradient dissimilarity cannot be satisfied for any $G \in \mathbb{R}$. □

In contrast, the $(G, B)$-gradient dissimilarity condition, presented in Assumption 1, is more realistic, since it allows the variance across the local gradients to grow with the norm of the global gradient. This condition is common in the (non-robust) distributed learning literature [23, 18, 25, 29], and is also well-known in the (non-distributed) optimization community [8, 9, 32]. While *G*-gradient dissimilarity corresponds to the special case of $(G, 0)$-gradient dissimilarity, we show in Proposition 1 below that a non-zero growth rate $B$ of gradient dissimilarity allows us to characterize heterogeneity in a much larger class of distributed learning problems.

**Proposition 1.** *Assume that the global loss $\mathcal{L}_\mathcal{H}$ is $\mu$-PL and L-smooth, and that for each $i \in \mathcal{H}$ local loss $\mathcal{L}_i$ is convex and $L_i$-smooth. Denote $L_{\max} := \max_{i \in \mathcal{H}} L_i$. Then, Assumption 1 is satisfied, i.e., the local loss functions satisfy $(G, B)$-gradient dissimilarity, with*

$$G^2 = \frac{2}{|\mathcal{H}|}\sum_{i \in \mathcal{H}} \|\nabla\mathcal{L}_i(\theta_*)\|^2 \qquad and \qquad B^2 = \frac{2L_{\max}}{\mu} - 1. \tag{4}$$

We present the proof of Proposition 1 in Appendix A for completeness. However, note that it can also be proved following existing results [32, 21] derived in other contexts. The $(G, B)$-gradient dissimilarity condition shown in Proposition 1 is arguably tight (up to multiplicative factor 2), in the sense that $G^2$ cannot be improved in general. Indeed, as the $(G, B)$-gradient dissimilarity inequality should be satisfied for $\theta = \theta_*$, $G^2$ should be at least the variance of honest gradients at the minimum, i.e., $\frac{1}{|\mathcal{H}|}\sum_{i \in \mathcal{H}} \|\nabla\mathcal{L}_i(\theta_*)\|^2$.

**Gap between existing theory and practice.** The theoretical limitation of *G*-gradient dissimilarity is exacerbated by the following empirical observation. We train a linear least-squares regression

model on the *mg* LIBSVM dataset [10]. The system comprises 7 honest and 3 Byzantine workers. We simulate *extreme heterogeneity* by having each honest worker hold one distinct point. We implement four well-studied Byzantine attacks: *sign flipping* (SF) [2], *fall of empires* (FOE) [33], *a little is enough* (ALIE) [5] and *mimic* [20]. More details on the experimental setup can be found in Appendix D. We consider the state-of-the-art robust variant of distributed gradient descent (detailed later in Section 5.1) that uses the NNM robustness scheme [3] composed with coordinate-wise trimmed mean. The empirical success on this learning task, which could not be explained by existing theory under $G$-gradient dissimilarity following Observation 1, is covered under $(G, B)$-gradient dissimilarity, as per Proposition 1. We present formal robustness guarantees under $(G, B)$-gradient dissimilarity later in Section 5. Additionally, through experimental evaluations in Section 5, we observe that even if $G$-gradient dissimilarity were assumed to be true, the bound $G^2$ may be extremely large in practice, thereby inducing a non-informative error bound $\mathcal{O}(f/n \cdot G^2)$ [20]. On the other hand, under $(G, B)$-gradient dissimilarity, we obtain tighter bounds matching empirical observations.

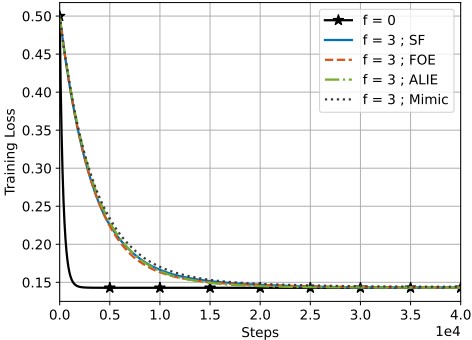 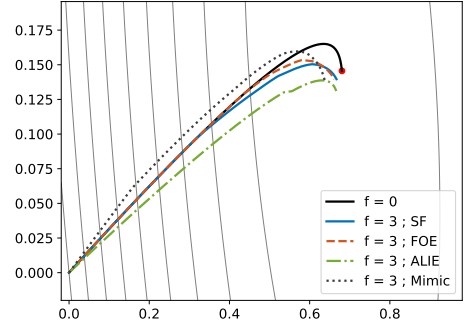

Figure 1: Evolution of the training losses **(left)** and the trajectories **(right)** of robust D-GD algorithm with NNM and coordinate-wise trimmed mean (see Section 5.1), on the *mg* LIBSVM least-squares regression task described in Section 3. $f = 0$ corresponds to the case where the algorithm is run without Byzantine workers.

## 4 Fundamental limits on robustness under $(G, B)$-Gradient Dissimilarity

Theorem 1 below shows the fundamental limits on robustness in distributed learning under $(G, B)$-gradient dissimilarity. The result has twofold implications. On the one hand, we show that the breakdown point of any robust distributed learning algorithm reduces with the growth rate $B$ of gradient dissimilarity. On the other hand, when the fraction $f/n$ is smaller than the breakdown point, both $G$ and $B$ induce a lower bound on the learning error.

**Theorem 1.** *Let $0 < f < n/2$. Assume that the global loss $\mathcal{L}_\mathcal{H}$ is $L$-smooth and $\mu$-strongly convex with $0 < \mu < L$. Assume also that the honest local losses satisfy $(G, B)$-gradient dissimilarity (Assumption 1) with $G > 0$. Then, a distributed algorithm can be $(f, \varepsilon)$-resilient only if*

$$\frac{f}{n} < \frac{1}{2 + B^2} \qquad and \qquad \varepsilon \geq \frac{1}{8\mu} \cdot \frac{f}{n - (2 + B^2) f} \cdot G^2 \,.$$

*Sketch of proof.* The full proof is deferred to Appendix B. In the proof, we construct hard instances for $(f, \varepsilon)$-resilience, using a set of quadratic functions of the following form:

$$\begin{aligned} \mathcal{L}_i(\theta) &= \tfrac{\alpha}{2} \|\theta - z\|^2 \quad , & \forall i \in \{1, \dots, f\}, \\ \mathcal{L}_i(\theta) &= \tfrac{1}{2K} \|\theta\|^2 \quad , & \forall i \in \{f + 1, \dots, n - f\}, \\ \mathcal{L}_i(\theta) &= \tfrac{\alpha}{2} \|\theta\|^2 \quad , & \forall i \in \{n - f + 1, \dots, n\}. \end{aligned}$$

To prove the theorem, we consider two plausible scenarios corresponding to two different identities of honest workers, which are *unknown* to the algorithm. Specifically, in scenarios I and II, we assume the indices of honest workers to be $S_1 := \{1, \dots, n - f\}$ and $S_2 := \{f + 1, \dots, n\}$, respectively.

We show that for all $\theta \in \mathbb{R}^d$,

$$\max \left\{ \mathcal{L}_{S_1}(\theta) - \mathcal{L}_{*,S_1}, \; \mathcal{L}_{S_2}(\theta) - \mathcal{L}_{*,S_2} \right\} \geq \frac{\left( \frac{f}{n-f} \right)^2 \alpha^2}{8 \left( \frac{n-2f}{n-f} \frac{1}{K} + \frac{f}{n-f} \alpha \right)} \left\| z \right\|^2 . \tag{5}$$

That is, every model in the parameter space incurs an error that grows with $\left\| z \right\|^2$, in at least one of the two scenarios. Hence, an $(f, \varepsilon)$-resilient algorithm, by Definition 1, must guarantee optimization error $\varepsilon$ in both scenarios I and II, which together with (5) implies that

$$\varepsilon \geq \frac{\left( \frac{f}{n-f} \right)^2 \alpha^2}{8 \left( \frac{n-2f}{n-f} \frac{1}{K} + \frac{f}{n-f} \alpha \right)} \left\| z \right\|^2 . \tag{6}$$

At this point, to obtain the largest lower bounds possible, our goal is to maximize the right-hand side of 6, under the constraint that the loss functions induced by the triplet $(\alpha, K, z)$ satisfy $(G, B)$-gradient dissimilarity, $L$-smoothness and $\mu$-strong convexity (simultaneously in both scenarios). We separately analyze this error in two cases: (i) $\frac{f}{n} \geq \frac{1}{2+B^2}$ and (ii) $\frac{f}{n} < \frac{1}{2+B^2}$. In both cases, we construct a coupling between the values of $z$ and $K$ by having the norm $\|z\|^2$ proportional to $K$. Specifically, in case (i), we show that the condition $\frac{f}{n} \geq \frac{1}{2+B^2}$ allows us to choose $K$ arbitrarily large while satisfying $(G, B)$-dissimilarity. Thus, $\|z\|^2$ being proportional to $K$ means that $\varepsilon$ is arbitrarily large as per (6). Similarly, in case (ii) where $\frac{f}{n} < \frac{1}{2+B^2}$, $K$ cannot be arbitrarily large and carefully choosing a large possible value yields

$$\varepsilon \geq \frac{1}{8\mu} \cdot \frac{f}{n - (2+B^2)f} \cdot G^2.$$

One of the crucial components to the above deductions was finding the suitable triplets $(\alpha, K, z)$ while preserving the $(G, B)$-gradient dissimilarity assumption (along with the smoothness and strong convexity assumptions) simultaneously in both the two scenarios, thereby establishing their validity. While the exact calculations are tedious, intuitively, $B$ constrains the relative difference between the scale parameters $\alpha$ and $\frac{1}{K}$, and $G$ constrains the separation between the minima, i.e. $\|z\|^2$. $\qquad \square$

**Extension to non-convex problems.** The lower bound from Theorem 1 assumes that the given distributed algorithm satisfies $(f, \varepsilon)$-resilience, which means finding an $\varepsilon$-approximate minimizer of the global honest loss $\mathcal{L}_{\mathcal{H}}$. The latter may not be possible for the general case of smooth and non-convex functions. In that case we cannot seek an $\varepsilon$-approximate minimizer, but rather an $\varepsilon$-approximate stationary point [20, 3], i.e., $\hat{\theta}$ such that $\|\nabla \mathcal{L}_{\mathcal{H}}(\hat{\theta})\|^2 \leq \varepsilon$. Then the lower bound in Theorem 1, in conjunction with the $\mu$-PL inequality, yields the following lower bound

$$\varepsilon \geq \frac{1}{4} \cdot \frac{f}{n - (2+B^2)f} \cdot G^2 . \tag{7}$$

**Comparison with prior work.** The result of Theorem 1 generalizes the existing robustness limits derived under $G$-gradient dissimilarity [20]. In particular, setting $B = 0$ in Theorem 1, we recover the breakdown point $\frac{1}{2}$ and the optimization lower bound $\Omega(f/n \cdot G^2)$. Perhaps, the most striking contrast to prior work [12, 20, 13, 3] is our breakdown point $\frac{1}{2+B^2}$, instead of simply $\frac{1}{2}$. We remark that a similar dependence on heterogeneity has been repeatedly assumed in the past, but without any formal justification. For instance, under $(0, B)$-gradient dissimilarity, [20, Theorem IV] assumes $f/n = \mathcal{O}(1/B^2)$ to obtain a formal robustness guarantee. In the context of robust distributed convex optimization (and robust least-squares regression), the upper bound assumed on the fraction $f/n$ usually depends upon the condition number of the distributed optimization problem, e.g., see [6, Theorem 3] and [16, Theorem 2]. Our analysis in Theorem 1 justifies these assumptions on the breakdown point in prior work under heterogeneity.

**Empirical breakdown point.** Interestingly, our breakdown point $\frac{1}{2+B^2}$ allows to better understand some empirical observations indicating that the breakdown point of robust distributed learning algorithms is smaller than $1/2$. We illustrate this in Figure 2 with a logistic regression model on the MNIST dataset under *extreme heterogeneity*, i.e., each worker dataset contains data points from a

single class. We consider the state-of-the-art robust variant of distributed gradient descent (detailed later in Section 5.1) that uses the NNM robustness scheme composed with robust aggregation rules, namely, *coordinate-wise trimmed-mean* (CW Trimmed Mean) [34], *Krum* [7], *coordinate-wise median* (CW Median) [34] and *geometric median* [31, 30, 1]. We observe that all these methods consistently fail to converge as soon as the fraction of Byzantine workers exceeds $\frac{1}{4}$, which is well short of the previously known theoretical breakdown point $\frac{1}{2}$. Theorem 1, to the best of our knowledge, provides the first formal justification to this empirical observation.

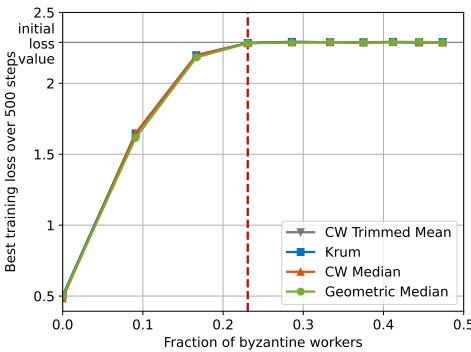 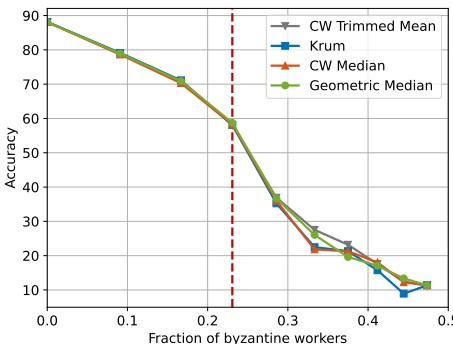

Figure 2: Best training loss **(left)** and accuracy **(right)** using robust D-GD (see Section 5.1) with NNM to train a logistic regression model on the MNIST dataset, in the presence of 10 honest workers and 1 to 9 Byzantine workers. The Byzantine workers use the sign flipping attack. More details on the experimental setup can be found in Appendix D.

## 5   Tight upper bounds under $(G, B)$-Gradient Dissimilarity

We demonstrate in this section that the bounds presented in Theorem 1 are tight. Specifically, we show that a robust variant of distributed gradient descent, referred to as *robust D-GD*, yields an asymptotic error that matches the lower bound under $(G, B)$-gradient dissimilarity, while also proving the tightness of the breakdown point. Lastly, we present empirical evaluations showcasing a significant improvement over existing robustness analyses that relied upon $G$-gradient dissimilarity.

### 5.1   Convergence analysis of robust D-GD

In robust D-GD, the server initially possesses a model $\theta_0$. Then, at each step $t \in \{0, \ldots, T-1\}$, the server broadcasts model $\theta_t$ to all workers. Each honest worker $w_i$ sends the gradient $g_t^{(i)} = \nabla \mathcal{L}_i(\theta_t)$ of its local loss function at $\theta_t$. However, a Byzantine worker $w_j$ might send an arbitrary value for its gradient. Upon receiving the gradients from all the workers, the server aggregates the local gradients using a *robust aggregation rule* $F \colon \mathbb{R}^{d \times n} \to \mathbb{R}^d$. Specifically, the server computes $R_t \coloneqq F(g_t^{(1)}, \ldots, g_t^{(n)})$. Ultimately, the server updates the current model $\theta_t$ to $\theta_{t+1} = \theta_t - \gamma R_t$, where $\gamma > 0$ is the *learning rate*. The full procedure is summarized in Algorithm 1.

---

**Algorithm 1:** Robust D-GD

**Input:** Initial model $\theta_0$, robust aggregation $F$, learning rate $\gamma$, and number of steps $T$.

**for** $t = 0 \ldots T - 1$ **do**

 **Server** broadcasts $\theta_t$ to all the workers;

 **for each** *honest worker* $\mathbf{w_i}$ **in parallel do**

  Compute and send gradient $g_t^{(i)} = \nabla \mathcal{L}_i(\theta_t)$;

  `// A Byzantine worker` $w_j$ `may send an arbitrary value for` $g_t^{(j)}$

 **Server** computes the aggregate gradient: $R_t = F(g_t^{(1)}, \ldots, g_t^{(n)})$;

 **Server** updates the model: $\theta_{t+1} = \theta_t - \gamma R_t$;

---

To analyze robust D-GD under $(G, B)$-gradient dissimilarity, we first recall the notion of $(f, \kappa)$-*robustness* in Definition 4 below. First introduced in [3], $(f, \kappa)$-robustness is a general property of robust aggregation that covers several existing aggregation rules.

**Definition 4** (($(f, \kappa)$-robustness). *Let $n \geq 1$, $0 \leq f < n/2$ and $\kappa \geq 0$. An aggregation rule $F \colon \mathbb{R}^{d \times n} \to \mathbb{R}^d$ is said to be $(f, \kappa)$-robust if for any vectors $x_1, \ldots, x_n \in \mathbb{R}^d$, and any set $S \subseteq [n]$ of size $n - f$, the output $\hat{x} := F(x_1, \ldots, x_n)$ satisfies the following:*

$$\|\hat{x} - \overline{x}_S\|^2 \leq \kappa \cdot \frac{1}{|S|} \sum_{i \in S} \|x_i - \overline{x}_S\|^2 \,,$$

*where $\overline{x}_S := \frac{1}{|S|} \sum_{i \in S} x_i$. We refer to $\kappa$ as the robustness coefficient of $F$.*

Closed-form robustness coefficients for multiple aggregation rules can be found in [3]. For example, assuming $n \geq (2 + \eta)f$, for some $\eta > 0$, $\kappa = \Theta(\frac{f}{n})$ for coordinate-wise trimmed mean, $\kappa = \Theta(1)$ for coordinate-wise median, and $\kappa_{F \circ \text{NNM}} = \Theta(\frac{f}{n}(\kappa + 1))$ when $F$ is composed with NNM [3].

Assuming $F$ to be an $(f, \kappa)$-robust aggregation rule, we show in Theorem 2 below the convergence of robust D-GD in the presence of up to $f$ Byzantine workers, under $(G, B)$-gradient dissimilarity.

**Theorem 2.** *Let $0 \leq f < n/2$. Assume that the global loss $\mathcal{L}_\mathcal{H}$ is $L$-smooth and that the honest local losses satisfy $(G, B)$-gradient dissimilarity (Assumption 1). Consider Algorithm 1 with learning rate $\gamma = \frac{1}{L}$. If the aggregation $F$ is $(f, \kappa)$-robust with $\kappa B^2 < 1$, then the following holds for all $T \geq 1$.*

1. *In the general case where $\mathcal{L}_\mathcal{H}$ may be non-convex, we have*

$$\frac{1}{T} \sum_{t=0}^{T-1} \|\nabla \mathcal{L}_\mathcal{H}(\theta_t)\|^2 \leq \frac{\kappa G^2}{1 - \kappa B^2} + \frac{2L\left(\mathcal{L}_\mathcal{H}(\theta_0) - \mathcal{L}_{*,\mathcal{H}}\right)}{(1 - \kappa B^2)T} \,.$$

2. *In the case where $\mathcal{L}_\mathcal{H}$ is $\mu$-PL, we have*

$$\mathcal{L}_\mathcal{H}(\theta_T) - \mathcal{L}_{*,\mathcal{H}} \leq \frac{\kappa G^2}{2\mu\left(1 - \kappa B^2\right)} + e^{-\frac{\mu}{L}\left(1 - \kappa B^2\right)T}\left(\mathcal{L}_\mathcal{H}(\theta_0) - \mathcal{L}_{*,\mathcal{H}}\right) \,.$$

**Tightness of the result.** We recall that the best possible robustness coefficient for an aggregation $F$ is $\kappa = f/n-2f$ (see [3]). For such an aggregation rule, the sufficient condition $\kappa B^2 < 1$ reduces to $f/n-2f \cdot B^2 < 1$ or equivalently $f/n < 1/2+B^2$. Besides, robust D-GD guarantees $(f, \varepsilon)$-resilience, for $\mu$-PL losses, where we have $\varepsilon = \mathcal{O}\left(f/n-(2+B^2)f \cdot G^2\right)$ asymptotically in $T$. Both these conditions on $f/n$ and $\varepsilon$ indeed match the limits shown earlier in Theorem 1. Yet, we are unaware of an aggregation rule with an order-optimal robustness coefficient, i.e., usually $\kappa > f/n-2f$. However, as shown in [3], the composition of *nearest neighbor mixing* (NNM) with several aggregation rules, such as CW Trimmed Mean, Krum and Geometric Median, yields a robustness coefficient $\kappa = \Theta(f/n-2f)$. Therefore, robust D-GD can indeed achieve $(f, \varepsilon)$-resilience with an optimal error $\varepsilon = \mathcal{O}\left(f/n-(2+B^2)f \cdot G^2\right)$, but for a suboptimal breakdown point. The same observation holds for the non-convex case, where the lower bound is given by (7). Lastly, note that when $B = 0$, our result recovers the bounds derived in prior work under $G$-gradient dissimilarity [20, 3].

We remark that while the convergence rate of robust D-GD shown in Theorem 2 is linear (which is typical to convergence of gradient descent in strongly convex case), it features a *slowdown factor* of value $1 - \kappa B^2$. Hence, suggesting that Byzantine workers might decelerate the training under heterogeneity. This slowdown is also empirically observed (e.g., see Figure 1). Whether this slowdown is fundamental to robust distributed learning is an interesting open question. Investigating such a slowdown in the stochastic case is also of interest, as existing convergence bounds are under $G$-gradient dissimilarity only, for strongly convex [4] and non-convex [3] cases.

**Comparison with prior work.** Few previous works have studied Byzantine robustness under $(G, B)$-gradient dissimilarity [20, 14]. While these works do not provide lower bounds, the upper bound they derive (see Appendix E in [20] and Appendix E.4 in [14]) are similar to Theorem 2, with some notable differences. First, unlike the notion of $(f, \kappa)$-robustness that we use, the so-called $(c, \delta)$-agnostic robustness, used in [20, 14], is a stochastic notion. Under the latter notion, good parameters $(c, \delta)$ of robust aggregators were only shown when using a randomized method called Bucketing [20]. Consequently, instead of obtaining a deterministic error bound as in Theorem 2,

simply replacing $c\delta$ with $\kappa$ in [20, 14] gives an expected bound, which is strictly weaker than the result of Theorem 2. Moreover, the corresponding non-vanishing upper bound term and breakdown point for robust D-GD obtained from the analysis in [20] for several robust aggregation rules (e.g., coordinate-wise median) are worse than what we obtain using $(f, \kappa)$-robustness.

## 5.2 Reducing the gap between theory and practice

In this section, we first argue that, even if we were to assume that the $G$-gradient dissimilarity condition (2) holds true, the robustness bounds derived in Theorem 2 under $(G, B)$-gradient dissimilarity improve upon the existing bounds [3] that rely on $G$-gradient dissimilarity. Next, we compare the empirical observations for robust D-GD with our theoretical upper bounds.

**Comparing upper bounds.** We consider a logistic regression model on MNIST dataset under extreme heterogeneity. While it is difficult to find tight values for parameters $G$ and $B$ satisfying $(G, B)$-gradient dissimilarity, we can approximate these parameters through a heuristic method. A similar approach can be used to approximate $\widehat{G}$ for which the loss functions satisfy the condition of $\widehat{G}$-gradient dissimilarity. We defer the details on these approximations to Appendix D. In Figure 3, we compare the error bounds, i.e., $f/n-(2+B^2)f \cdot G^2$ and $f/n-2f \cdot \widehat{G}^2$, guaranteed for robust D-GD under $(G, B)$-gradient dissimilarity and $\widehat{G}$-gradient dissimilarity, respectively. We observe that the latter bound is extremely large compared to the former, which confirms that the tightest bounds under $G$-gradient dissimilarity are vacuous for practical purposes.

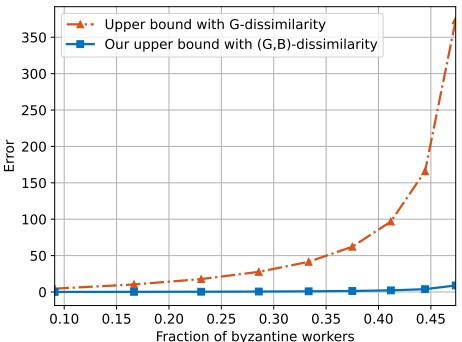

Figure 3: Comparison of our upper bound in Theorem 2 with that of $G$-gradient dissimilarity on MNIST with a logistic regression model. The number of honest workers is 10, and the number of Byzantine workers varies from 1 to 9.

Figure 4: Comparison between our upper bound in Corollary 1 and the training loss for the least-squares regression on the *mg* LIBSVM dataset [10]. The number of honest workers is 20 and the number of Byzantine workers varies from 1 to 19.

We further specialize the result of Theorem 2 to the convex case for which the $(G, B)$-gradient dissimilarity condition was characterized in Proposition 1. We have the following corollary.

**Corollary 1.** *Assume that the global loss $\mathcal{L}_{\mathcal{H}}$ is $\mu$-PL and $L$-smooth, and that for each $i \in \mathcal{H}$ local loss $\mathcal{L}_i$ is convex and $L_i$-smooth. Denote $L_{\max} := \max_{i \in \mathcal{H}} L_i$ and assume that $\kappa\left(\frac{3L_{\max}}{\mu} - 1\right) \le 1$. Consider Algorithm 1 with learning rate $\gamma = \frac{1}{L}$. If $F$ is $(f, \kappa)$-robust, then for all $T \ge 1$, we have*

$$\mathcal{L}_{\mathcal{H}}(\theta_T) - \mathcal{L}_{*,\mathcal{H}} \le \frac{3\kappa}{\mu} \frac{1}{|\mathcal{H}|} \sum_{i \in \mathcal{H}} \|\nabla \mathcal{L}_i(\theta_*)\|^2 + e^{-\frac{\mu}{3L}T} \left(\mathcal{L}_{\mathcal{H}}(\theta_0) - \mathcal{L}_{*,\mathcal{H}}\right).$$

The non-vanishing term in the upper bound shown in Corollary 1 corresponds to the heterogeneity at the minimum $\frac{1}{|\mathcal{H}|} \sum_{i \in \mathcal{H}} \|\nabla \mathcal{L}_i(\theta_*)\|^2$. This quantity is considered to be a natural measure of gradient dissimilarity in classical (non-Byzantine) distributed convex optimization [22, 23]. As such, we believe that this bound cannot be improved upon in general.

**Matching empirical performances.** Since the upper bound in Corollary 1 requires computing the constants $\mu, L$, we choose to conduct this experiment on least-squares regression, where the exact

computation of $\mu, L$ is possible. We compare the empirical error gap (left-hand side of Corollary 1) with the upper bound (right-hand side of Corollary 1). Our findings, shown in Figure 4, indicate that our theoretical analysis reliably predicts the empirical performances of robust D-GD, especially when the fraction of Byzantine workers is small. Note, however, that our upper bound is non-informative when more than $\frac{1}{4}$ of the workers are Byzantine, as the predicted error exceeds the initial loss value. We believe this to be an artifact of the proof, i.e. the upper bound is meaningful only up to a multiplicative constant. Indeed, when visualizing the results in logarithmic scale in Figure 4, the shape of empirical measurements and our upper bounds are quite similar.

# 6 Conclusion and future work

This paper revisits the theory of robust distributed learning by considering a realistic data heterogeneity model, namely $(G, B)$-gradient dissimilarity. Using this model, we show that the breakdown point depends upon heterogeneity (specifically, $1/2+B^2$) and is smaller than the usual fraction $1/2$. We prove a new lower bound on the learning error of any distributed learning algorithm, which is matched using robust D-GD. Moreover, we show that our theoretical guarantees align closely with empirical observations, contrary to prior works which rely upon the stringent model of $G$-gradient dissimilarity.

An interesting future research direction is to investigate whether the $1 - \kappa B^2$ slowdown factor in the convergence rate of robust D-GD (Theorem 2) is unavoidable. Another interesting research problem is to derive lower (and upper) bounds independent of the heterogeneity model, thereby elucidating the tightness of the convergence guarantee of robust D-GD in the strongly convex case (Corollary 1).

## Acknowledgements

This work was supported in part by SNSF grant 200021_200477 and an EPFL-INRIA postdoctoral grant. The authors are thankful to the anonymous reviewers for their constructive comments.

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

## Organization of the Appendix

Appendix A contains the proof of Proposition 1. Appendix B contains the impossibility result shown in Theorem 1. Appendix C contains the convergence proofs concerning Robust D-GD (Theorem 2, Corollary 1, and Proposition 1). Appendix D presents our experimental setups.

## A    Proof of Proposition 1

**Proposition 1.** *Assume that the global loss $\mathcal{L}_{\mathcal{H}}$ is $\mu$-PL and $L$-smooth, and that for each $i \in \mathcal{H}$ local loss $\mathcal{L}_i$ is convex and $L_i$-smooth. Denote $L_{\max} := \max_{i \in \mathcal{H}} L_i$. Then, Assumption 1 is satisfied, i.e., the local loss functions satisfy $(G, B)$-gradient dissimilarity, with*

$$G^2 = \frac{2}{|\mathcal{H}|} \sum_{i \in \mathcal{H}} \|\nabla \mathcal{L}_i(\theta_*)\|^2 \qquad and \qquad B^2 = \frac{2L_{\max}}{\mu} - 1. \tag{4}$$

*Proof.* Let $\theta \in \mathbb{R}^d$. By bias-variance decomposition, we obtain that

$$\frac{1}{|\mathcal{H}|} \sum_{i \in \mathcal{H}} \|\nabla \mathcal{L}_i(\theta) - \nabla \mathcal{L}_{\mathcal{H}}(\theta)\|^2 = \frac{1}{|\mathcal{H}|} \sum_{i \in \mathcal{H}} \|\nabla \mathcal{L}_i(\theta)\|^2 - \|\nabla \mathcal{L}_{\mathcal{H}}(\theta)\|^2. \tag{8}$$

Using triangle inequality, we have

$$\|\nabla \mathcal{L}_i(\theta)\|^2 = \|\nabla \mathcal{L}_i(\theta) - \nabla \mathcal{L}_i(\theta_*) + \nabla \mathcal{L}_i(\theta_*)\|^2 \leq (\|\nabla \mathcal{L}_i(\theta) - \nabla \mathcal{L}_i(\theta_*)\| + \|\nabla \mathcal{L}_i(\theta_*)\|)^2$$

For any pair of real values $(a, b)$, we have $(a + b)^2 \leq 2a^2 + 2b^2$. Using this inequality with $a = \|\nabla \mathcal{L}_i(\theta) - \nabla \mathcal{L}_i(\theta_*)\|$ and $b = \|\nabla \mathcal{L}_i(\theta_*)\|$ in the above, we obtain that

$$\|\nabla \mathcal{L}_i(\theta)\|^2 \leq 2 \|\nabla \mathcal{L}_i(\theta) - \nabla \mathcal{L}_i(\theta_*)\|^2 + 2 \|\nabla \mathcal{L}_i(\theta_*)\|^2.$$

Using the above in (8), we obtain that

$$\frac{1}{|\mathcal{H}|} \sum_{i \in \mathcal{H}} \|\nabla \mathcal{L}_i(\theta) - \nabla \mathcal{L}_{\mathcal{H}}(\theta)\|^2 \leq \frac{1}{|\mathcal{H}|} \sum_{i \in \mathcal{H}} \left( 2 \|\nabla \mathcal{L}_i(\theta) - \nabla \mathcal{L}_i(\theta_*)\|^2 + 2 \|\nabla \mathcal{L}_i(\theta_*)\|^2 \right)$$
$$- \|\nabla \mathcal{L}_{\mathcal{H}}(\theta)\|^2. \tag{9}$$

For all $i \in \mathcal{H}$, since $\mathcal{L}_i$ is assumed convex and $L_i$-smooth, we also have (see [28, Theorem 2.1.5]) for all $\theta' \in \mathbb{R}^d$ that

$$\mathcal{L}_i(\theta) \geq \mathcal{L}_i(\theta') + \langle \nabla \mathcal{L}_i(\theta'), \theta - \theta' \rangle + \frac{1}{2L_i} \|\nabla \mathcal{L}_i(\theta) - \nabla \mathcal{L}_i(\theta')\|^2.$$

Substituting $\theta' = \theta_*$ in the above, we have $\mathcal{L}_i(\theta) - \mathcal{L}_i(\theta_*) - \langle \nabla \mathcal{L}_i(\theta_*), \theta - \theta_* \rangle \geq 0$ and $\|\nabla \mathcal{L}_i(\theta) - \nabla \mathcal{L}_i(\theta_*)\|^2 \leq 2L_i (\mathcal{L}_i(\theta) - \mathcal{L}_i(\theta_*) - \langle \nabla \mathcal{L}_i(\theta_*), \theta - \theta_* \rangle)$. Therefore, as $L_{\max} := \max_{i \in \mathcal{H}} L_i$, we have

$$\frac{1}{|\mathcal{H}|} \sum_{i \in \mathcal{H}} \|\nabla \mathcal{L}_i(\theta) - \nabla \mathcal{L}_i(\theta_*)\|^2 \leq \frac{2L_{\max}}{|\mathcal{H}|} \sum_{i \in \mathcal{H}} (\mathcal{L}_i(\theta) - \mathcal{L}_i(\theta_*) - \langle \nabla \mathcal{L}_i(\theta_*), \theta - \theta_* \rangle). \tag{10}$$

Recall that $\mathcal{L}_{\mathcal{H}} := \frac{1}{|\mathcal{H}|} \sum_{i \in \mathcal{H}} \mathcal{L}_i$. Thus, $\frac{1}{|\mathcal{H}|} \sum_{i \in \mathcal{H}} \mathcal{L}_i(\theta_*) = \mathcal{L}_{\mathcal{H}}(\theta_*)$, and $\frac{1}{|\mathcal{H}|} \sum_{i \in \mathcal{H}} \nabla \mathcal{L}_i(\theta_*) = \nabla \mathcal{L}_{\mathcal{H}}(\theta_*) = 0$. Using this in (10), and then recalling that $\mathcal{L}_{\mathcal{H}}$ is assumed $\mu$-PL, we have

$$\frac{1}{|\mathcal{H}|} \sum_{i \in \mathcal{H}} \|\nabla \mathcal{L}_i(\theta) - \nabla \mathcal{L}_i(\theta_*)\|^2 \leq 2L_{\max} (\mathcal{L}_{\mathcal{H}}(\theta) - \mathcal{L}_{\mathcal{H}}(\theta_*)) \leq \frac{L_{\max}}{\mu} \|\nabla \mathcal{L}_{\mathcal{H}}(\theta)\|^2.$$

Substituting the above in (9), we obtain that

$$\frac{1}{|\mathcal{H}|} \sum_{i \in \mathcal{H}} \|\nabla \mathcal{L}_i(\theta) - \nabla \mathcal{L}_{\mathcal{H}}(\theta)\|^2 \leq 2 \frac{L_{\max}}{\mu} \|\nabla \mathcal{L}_{\mathcal{H}}(\theta)\|^2 + 2 \frac{1}{|\mathcal{H}|} \sum_{i \in \mathcal{H}} \|\nabla \mathcal{L}_i(\theta_*)\|^2 - \|\nabla \mathcal{L}_{\mathcal{H}}(\theta)\|^2$$
$$= \frac{2}{|\mathcal{H}|} \sum_{i \in \mathcal{H}} \|\nabla \mathcal{L}_i(\theta_*)\|^2 + \left( \frac{2L_{\max}}{\mu} - 1 \right) \|\nabla \mathcal{L}_{\mathcal{H}}(\theta)\|^2.$$

The above proves the proposition. $\qquad\square$

# B  Proof of Theorem 1: Impossibility Result

For convenience, we recall below the theorem statement.

**Theorem 1.** *Let $0 < f < n/2$. Assume that the global loss $\mathcal{L}_\mathcal{H}$ is L-smooth and $\mu$-strongly convex with $0 < \mu < L$. Assume also that the honest local losses satisfy $(G, B)$-gradient dissimilarity (Assumption 1) with $G > 0$. Then, a distributed algorithm can be $(f, \varepsilon)$-resilient only if*

$$\frac{f}{n} < \frac{1}{2 + B^2} \qquad and \qquad \varepsilon \geq \frac{1}{8\mu} \cdot \frac{f}{n - (2 + B^2)f} \cdot G^2 .$$

## B.1  Proof outline

We prove the theorem by contradiction. We start by assuming that there exists an algorithm $\mathcal{A}$ that is $(f, \varepsilon)$-resilient when the conditions stated in the theorem for the honest workers are satisfied. We consider the following instance of the loss functions where parameters $\alpha$ and $K$ are positive real values, and $z \in \mathbb{R}^d$ is a vector.

$$\mathcal{L}_i(\theta) = \mathcal{L}_\mathrm{I}(\theta) := \frac{\alpha}{2} \|\theta - z\|^2 , \qquad\qquad \forall i \in \{1, \ldots, f\}, \qquad (11)$$

$$\mathcal{L}_i(\theta) = \mathcal{L}_\mathrm{II}(\theta) := \frac{1}{2K} \|\theta\|^2 , \qquad\qquad \forall i \in \{f + 1, \ldots, n - f\}, \qquad (12)$$

$$\mathcal{L}_i(\theta) = \mathcal{L}_\mathrm{III}(\theta) := \frac{\alpha}{2} \|\theta\|^2 , \qquad\qquad \forall i \in \{n - f + 1, \ldots, n\}. \qquad (13)$$

We then consider two specific scenarios, each corresponding to different identities of honest workers: $S_1 = \{1, \ldots, n - f\}$ and $S_2 = \{f + 1, \ldots, n\}$. That is, we let $S_1$ and $S_2$ represent the set of honest workers in the first and second scenarios, respectively. Upon specifying certain conditions on parameters $\alpha$, $K$ and $z$ we show that the corresponding honest workers' loss functions in either execution satisfy the assumptions stated in the theorem, i.e., the global loss functions are $L$-smooth $\mu$-strongly convex and the honest local loss functions satisfy $(G, B)$-gradient dissimilarity. Since algorithm $\mathcal{A}$ is oblivious to the honest identities, it must ensure $(f, \varepsilon)$-resilience in both these scenarios. Consequently, we show that $\varepsilon$ cannot be lower that a value that grows with $K$ and $\|z\|^2$. Using this approach, we first show that the lower bound on $\varepsilon$ can be arbitrarily large when $\frac{f}{n} \geq \frac{1}{2+B^2}$. Then, we obtain a non-trivial lower bound on $\varepsilon$ in the case when $\frac{f}{n} < \frac{1}{2+B^2}$. The two scenarios, and corresponding loss functions, used in our proof are illustrated in Figure 5.

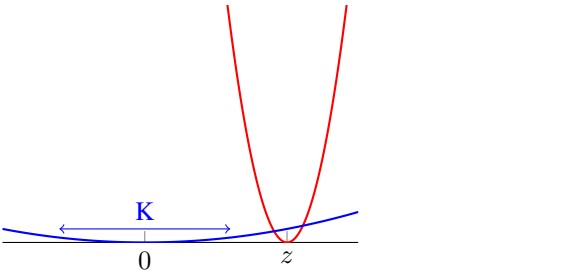 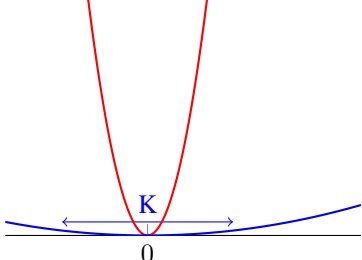

Figure 5: Illustration of the proof of Theorem 1. It is impossible to distinguish between the two scenarios depicted above, corresponding to the local honest losses in scenarios $S_1$ and $S_2$. We set $\|z\|^2$ to grow with $K$, and show that $(G, B)$-gradient dissimilarity holds in both scenarios. A large $K$ means that the minimum in the first scenario (left) is close to $z$, while it is 0 in the second scenario (right). Thus, any algorithm must make an error $\varepsilon$ in the order of $\|z\|^2$, which itself grows with $K$. When $f/n \geq \frac{1}{2+B^2}$, we show that $K$ and thus $\varepsilon$ can be made arbitrarily large.

We make use of the following auxiliary results.

### B.1.1  Unavoidable error due to anonymity of Byzantine workers

Lemma 1 below establishes a lower bound on the optimization error that any algorithm must incur in at least one of the two scenarios described above. Recall that $S_1 = \{1, \ldots, n - f\}$ and

$S_2 = \{f+1, \ldots, n\}$. Recall that for any non-empty subset $S \subseteq [n]$, we denote

$$\mathcal{L}_S(\theta) := \frac{1}{|S|} \sum_{i \in S} \mathcal{L}_i(\theta) \quad \text{and} \quad \mathcal{L}_{*,S} = \min_{\theta \in \mathbb{R}^d} \mathcal{L}_S(\theta) \,.$$

**Lemma 1.** *Consider the setting where the local loss functions are given by* (11), (12), *and* (13). *In this particular case, the following holds for all $\theta \in \mathbb{R}^d$:*

$$\max \left\{ \mathcal{L}_{S_1}(\theta) - \mathcal{L}_{*,S_1}, \ \mathcal{L}_{S_2}(\theta) - \mathcal{L}_{*,S_2} \right\} \geq \frac{\left( \frac{f}{n-f} \right)^2 \alpha^2}{8 \left( \frac{n-2f}{n-f} \frac{1}{K} + \frac{f}{n-f} \alpha \right)} \|z\|^2 \,.$$

The proof is deferred to Appendix B.3. We next analyze the $(G, B)$-gradient dissimilarity condition for the considered distributed learning setting.

### B.1.2 Validity of $(G, B)$-gradient dissimilarity

In Lemma 2 below we derive necessary and sufficient conditions on $\alpha$, $K$ and $z$ for $(G, B)$-gradient dissimilarity when the loss functions are given by (11), (12) and (13).

**Lemma 2.** *Consider the setting where the local loss functions are given by* (11), (12), *and* (13). *Denote*

$$A_1 := \frac{f(n-2f)}{(n-f)^2} \left( \left( 1 - \frac{n-2f}{f} B^2 \right) \frac{1}{K^2} - 2(1+B^2)\frac{\alpha}{K} + \left( 1 - \frac{f}{n-2f} B^2 \right) \alpha^2 \right) \,,$$

$$A_2 := \frac{f(n-2f)\alpha}{(n-f)((n-2f)\frac{1}{K}+f\alpha)} \left( \frac{1}{K} - \alpha \right) \frac{1}{K} z \,, \text{ and } A_3 := \frac{f(n-2f)\alpha^2}{\left( (n-2f)\frac{1}{K}+f\alpha \right)^2} \frac{\|z\|^2}{K^2} \,. \tag{14}$$

*Suppose that $S_1 = \{1, \ldots, n-f\}$ denotes the set of honest workers. Then, the honest workers satisfy $(G, B)$-gradient dissimilarity if and only if*

$$A_1 \leq 0, \qquad A_3 \leq G^2, \quad \text{and} \quad \|A_2\|^2 \leq A_1(A_3 - G^2). \tag{15}$$

The proof of Lemma 2 is deferred to Appendix B.4. Since $\mathcal{L}_{\mathrm{I}}$ corresponds to $\mathcal{L}_{\mathrm{III}}$ with $z = 0$, the result in the lemma above also holds true when the honest workers is represented by set $S_2 = \{f+1, \ldots, n\}$. Specifically, we have the following lemma.

**Lemma 3.** *Consider a specific distributed learning setting where the local loss functions are as defined in* (11), (12) *and* (13). *We denote*

$$A_1 := \frac{f(n-2f)}{(n-f)^2} \left( \left( 1 - \frac{n-2f}{f} B^2 \right) \frac{1}{K^2} - 2(1+B^2)\frac{\alpha}{K} + \left( 1 - \frac{f}{n-2f} B^2 \right) \alpha^2 \right) \,.$$

*Suppose that $S_2 = \{f+1, \ldots, n\}$ denotes the set of honest workers. Then, the honest workers satisfy $(0, B)$-gradient dissimilarity if and only if $A_1 \leq 0$.*

We do not provide a proof of Lemma 3, as it follows the proof of Lemma 2 verbatim upon simply substituting $z = 0_{\mathbb{R}^d}$ and $G = 0$.

### B.2 Proof of Theorem 1

We prove the two assertions of Theorem 1, i.e., the necessity of $\frac{f}{n} < \frac{1}{2+B^2}$ and the lower bound on $\varepsilon$, separately in sections B.2.1 and B.2.2, respectively.

### B.2.1 Necessity of $\frac{f}{n} < \frac{1}{2+B^2}$

In this section, we prove by contradiction the necessity of $\frac{f}{n} < \frac{1}{2+B^2}$ by demonstrating that $\varepsilon$ can be arbitrarily large if $\frac{f}{n} \geq \frac{1}{2+B^2}$. Let $0 < f < n/2$, $0 < \mu < L$, $G > 0$ and $B \geq 0$.

Suppose that $\frac{f}{n} \geq \frac{1}{2+B^2}$, or equivalently $B^2 \geq \frac{n-2f}{f}$. Let $\mathcal{A}$ be an arbitrary $(f, \varepsilon)$-resilient distributed learning algorithm. We consider the setting where the local loss functions are given by (11), (12) and (13), and the corresponding parameters are given by

$$\alpha = \frac{n-f}{f} \mu \,, \tag{16}$$

$K$ is an arbitrary positive real number such that

$$K \geq \frac{1}{\alpha} \max \left\{ 1 \,, \frac{n-2f}{f} \cdot \frac{\mu}{L-\mu} \right\} \,, \tag{17}$$

and $z \in \mathbb{R}^d$ is such that

$$\|z\|^2 = \frac{f}{n-2f} \cdot \frac{G^2}{2\alpha} K \,. \tag{18}$$

**Proof outline.** In the proof, we consider two scenarios, each corresponding to two different identities of honest workers: $S_1 = \{1, \ldots, n-f\}$ and $S_2 = \{f+1, \ldots, n\}$. For each of these, we first show that the corresponding local and global honest loss functions satisfy the assumptions made in the theorem, invoking Lemma 2. Then, by invoking Lemma 1, we show that $\varepsilon$ is proportional to $K$ which (as per (17)) can be chosen to be arbitrarily large. This yields a contradiction to the assumption that $\mathcal{A}$ is $(f, \varepsilon)$-resilient, proving that $(f, \varepsilon)$-resilience is generally impossible when $\frac{f}{n} \geq \frac{1}{2+B^2}$.

**First scenario.** Suppose that the set of honest workers is represented by $S_1 = \{1, \ldots, n-f\}$. From (11) and (12), we obtain that

$$\mathcal{L}_{S_1}(\theta) = \frac{1}{|S_1|} \sum_{i \in S_1} \mathcal{L}_i(\theta) = \frac{f}{n-f} \mathcal{L}_{\mathrm{I}}(\theta) + \frac{n-2f}{n-f} \mathcal{L}_{\mathrm{II}}(\theta)$$

$$= \frac{f}{n-f} \frac{\alpha}{2} \|\theta - z\|^2 + \frac{n-2f}{n-f} \frac{1}{2K} \|\theta\|^2 \,.$$

Substituting, from (16), $\alpha = \frac{n-f}{f} \mu$ in the above we obtain that

$$\mathcal{L}_{S_1}(\theta) = \frac{\mu}{2} \|\theta - z\|^2 + \frac{n-2f}{n-f} \frac{1}{2K} \|\theta\|^2 \,. \tag{19}$$

Therefore,

$$\nabla \mathcal{L}_{S_1}(\theta) = \mu (\theta - z) + \frac{n-2f}{n-f} \frac{1}{K} \theta \,, \text{ and } \nabla^2 \mathcal{L}_{S_1}(\theta) = \left( \mu + \frac{n-2f}{n-f} \frac{1}{K} \right) I_d \,, \tag{20}$$

where $I_d$ denotes the identity matrix of size $d$. From (20), we deduce that $\mathcal{L}_{S_1}(\theta)$ is $\left( \frac{n-2f}{n-f} \frac{1}{K} + \mu \right)$-smooth and $\left( \frac{n-2f}{n-f} \frac{1}{K} + \mu \right)$-strongly convex. From (17), we have $K \geq \frac{1}{\alpha} \cdot \frac{n-2f}{f} \cdot \frac{\mu}{L-\mu} = \frac{n-2f}{n-f} \cdot \frac{1}{L-\mu}$. This implies that $\frac{n-2f}{n-f} \frac{1}{K} + \mu \leq L - \mu + \mu = L$. Clearly, $\left( \frac{n-2f}{n-f} \frac{1}{K} + \mu \right) \geq \mu$. Hence, the honest global loss function $\mathcal{L}_{S_1}$ is $L$-smooth $\mu$-strongly convex. Next, invoking Lemma 2, we show that the honest workers satisfy $(G, B)$-gradient dissimilarity. We analyze below the terms $A_1, A_2$ and $A_3$ introduced in Lemma 2 in this particular scenario.

*Term $A_1$.* Recall from Lemma 2 that

$$A_1 = \frac{f(n-2f)}{(n-f)^2} \left( \left( 1 - \frac{n-2f}{f} B^2 \right) \frac{1}{K^2} - 2(1+B^2) \frac{\alpha}{K} + \alpha^2 \left( 1 - \frac{f}{n-2f} B^2 \right) \right) \,.$$

Since we assume $B^2 \geq \frac{n-2f}{f}$, we have $1 - \frac{f}{n-2f} B^2 \leq 0$. Using this in the above we obtain that

$$A_1 \leq \frac{f(n-2f)}{(n-f)^2} \left( \left( 1 - \frac{n-2f}{f} B^2 \right) \frac{1}{K^2} - 2(1+B^2) \frac{\alpha}{K} \right)$$

$$= \frac{f(n-2f)}{(n-f)^2} \frac{\alpha}{K} \left( \left( 1 - \frac{n-2f}{f} B^2 \right) \frac{1}{\alpha K} - 2(1+B^2) \right) \,.$$

As $\alpha, K > 0$, and by (17), $K \geq \frac{1}{\alpha} \geq \frac{1}{\alpha} \frac{\left(1 - \frac{n-2f}{f}B^2\right)}{1+B^2}$. Therefore, $1 + B^2 \geq \frac{1}{\alpha K}\left(1 - \frac{n-2f}{f}B^2\right)$. Using this in the above implies that

$$A_1 \leq -\frac{f(n-2f)}{(n-f)^2} \frac{\alpha}{K}(1 + B^2) \leq -\frac{f(n-2f)}{(n-f)^2} \frac{\alpha}{K}. \tag{21}$$

*Term $A_2$.* Recall from Lemma 2 that

$$A_2 = \frac{f(n-2f)\alpha}{(n-f)((n-2f)\frac{1}{K} + f\alpha)}\left(\frac{1}{K} - \alpha\right)\frac{1}{K}z. \tag{22}$$

Therefore, as $K > 0$ and we assume $n > 2f$, we have

$$\|A_2\|^2 = \left(\frac{f(n-2f)\alpha}{(n-f)((n-2f)\frac{1}{K} + f\alpha)}\right)^2 \left(\frac{1}{K} - \alpha\right)^2 \frac{1}{K^2}\|z\|^2$$

$$\leq \left(\frac{f(n-2f)\alpha}{(n-f)f\alpha}\right)^2 \left(\frac{1}{K} - \alpha\right)^2 \frac{1}{K^2}\|z\|^2 = \left(\frac{n-2f}{n-f}\right)^2 \left(\alpha - \frac{1}{K}\right)^2 \frac{1}{K^2}\|z\|^2.$$

As $K \geq \frac{1}{\alpha} > 0$, we have $\left(\alpha - \frac{1}{K}\right)^2 \leq \alpha^2$. Thus, from above we obtain that

$$\|A_2\|^2 \leq \left(\frac{n-2f}{n-f}\right)^2 \alpha^2 \frac{1}{K^2}\|z\|^2,$$

Substituting from (18), i.e. $\|z\|^2 = \frac{f}{n-2f}\frac{G^2}{2\alpha} \cdot K$, in the above implies that

$$\|A_2\|^2 \leq \frac{f(n-2f)}{(n-f)^2}\frac{\alpha}{K}\frac{G^2}{2}. \tag{23}$$

*Term $A_3$.* Recall from Lemma 2 that

$$A_3 := \frac{f(n-2f)\alpha^2}{\left((n-2f)\frac{1}{K} + f\alpha\right)^2}\frac{\|z\|^2}{K^2}.$$

As $K > 0$ and $n > 2f$, we have

$$A_3 \leq \frac{f(n-2f)\alpha^2}{f^2\alpha^2}\frac{\|z\|^2}{K^2} = \frac{n-2f}{f}\frac{\|z\|^2}{K^2}.$$

Substituting $\|z\|^2 = \frac{f}{n-2f}\frac{G^2}{2\alpha} \cdot K$ (see (18)), and then recalling that $K \geq \frac{1}{\alpha}$ (see (17)), yields

$$A_3 \leq \frac{G^2}{2\alpha K} \leq \frac{G^2}{2}.$$

Therefore,

$$G^2 - A_3 \geq \frac{G^2}{2} \geq 0. \tag{24}$$

*Invoking Lemma 2.* Using the results obtained above in (21), (23) and (24), we show below that the conditions stated in (15) of Lemma (2) are satisfied, i.e., $A_1 \leq 0$, $A_3 \leq G^2$ and $\|A_2\|^2 \leq A_1(A_3 - G^2)$. Hence, we prove that the local loss functions for the honest workers in this particular scenario satisfy $(G, B)$-gradient dissimilarity.

First, from (21), we note that

$$A_1 \leq -\frac{f(n-2f)}{(n-f)^2}\frac{\alpha}{K} \leq 0. \tag{25}$$

Second, as a consequence of (24), $A_3 \leq G^2$. Lastly, recall from (23) that

$$\|A_2\|^2 \leq \frac{f(n-2f)}{(n-f)^2}\frac{\alpha}{K}\frac{G^2}{2}.$$

As $\frac{f(n-2f)}{(n-f)^2}\frac{\alpha}{K} \leq -A_1$ (from (21)), from above we obtain that $\|A_2\|^2 \leq -A_1\frac{G^2}{2}$. Therefore, as $G^2 \geq 0$ and $G^2 \leq 2(G^2 - A_3)$ (from (24)), $\|A_2\|^2 \leq A_1(A_3 - G^2)$.

This concludes the analysis for the first scenario. We have shown that the conditions on smoothness, strong convexity and $(G, B)$-gradient dissimilarity hold in this scenario.

**Second scenario.** Consider the set of honest workers to be $S_2 = \{f + 1, \ldots, n\}$. Identical to (19) with $z = 0$, from (12) and (13), we obtain that

$$\mathcal{L}_{S_2}(\theta) = \left(\frac{\mu}{2} + \frac{n - 2f}{n - f}\frac{1}{2K}\right)\|\theta\|^2 .$$

Therefore, similar to the analysis of the first scenario, the global loss $\mathcal{L}_{S_2}$ satisfies $L$-smoothness and $\mu$-strong convexity. Moreover, Lemma 3, in conjunction with the deduction in (25) that $A_1 \geq 0$, implies that the loss functions for the honest workers in this particular scenario satisfy $(0, B)$-gradient dissimilarity, thereby also satisfying $(G, B)$-gradient dissimilarity.

This concludes the analysis for the second scenario. We have shown that the conditions on smoothness, strong convexity and $(G, B)$-gradient dissimilarity indeed hold true in this scenario.

**Final step: lower bound on $\varepsilon$ in terms of $K$.** We have established in the above that the conditions of smoothness, strong convexity and $(G, B)$-gradient dissimilarity are satisfied in both the scenarios. Therefore, by assumptions, algorithm $\mathcal{A}$ must guarantee $(f, \varepsilon)$-resilience in either scenario. Specifically, the output of $\mathcal{A}$, denoted by $\hat{\theta}$ must satisfy the following.

$$\max\left\{\mathcal{L}_{S_1}(\hat{\theta}) - \mathcal{L}_{*,S_1} , \ \mathcal{L}_{S_2}(\hat{\theta}) - \mathcal{L}_{*,S_2}\right\} \leq \varepsilon.$$

Due to Lemma 1, the above holds true only if

$$\varepsilon \geq \frac{\left(\frac{f}{n-f}\right)^2\alpha^2}{8\left(\frac{n-2f}{n-f}\frac{1}{K} + \frac{f}{n-f}\alpha\right)}\|z\|^2 . \tag{26}$$

Recall from (17) that $K \geq \frac{1}{\alpha} \cdot \frac{n-2f}{f} \cdot \frac{\mu}{L-\mu}$. Therefore, we have $(n - 2f)\frac{1}{K} \leq \frac{L-\mu}{\mu}f\alpha$, which implies that

$$\frac{\left(\frac{f}{n-f}\right)^2\alpha^2}{\frac{n-2f}{n-f}\frac{1}{K} + \frac{f}{n-f}\alpha} \geq \frac{\left(\frac{f}{n-f}\right)^2\alpha^2}{\left(\frac{L-\mu}{\mu} + 1\right)\frac{f}{n-f}\alpha} = \frac{\mu}{L} \cdot \frac{f}{n-f}\alpha.$$

Using the above in (26), and then substituting $\alpha = \frac{n-f}{f}\mu$ (from (16)), implies that

$$\varepsilon \geq \frac{\mu^2}{8L}\|z\|^2 .$$

Recall from (18) that $\|z\|^2 = \frac{f}{n-2f}\frac{G^2}{2\alpha} \cdot K$ where $\alpha = \frac{n-f}{f}\mu$, from above we obtain that

$$\varepsilon \geq \frac{\mu^2}{16L}\frac{f}{n-2f}\frac{G^2}{\alpha}K = \frac{\mu}{16L} \cdot \frac{f^2}{(n-2f)(n-f)}G^2K.$$

That is, $\varepsilon$ grows with $K$. Note that the above holds for any arbitrarily large value of $K$ satisfying (17). Therefore, $\varepsilon$ can be made arbitrarily large. This contracts the assumption that $\mathcal{A}$ is $(f, \varepsilon)$-resilient with a finite $\varepsilon$. Hence, we have shown that $(f, \varepsilon)$-resilience is impossible in general under $(G, B)$-gradient dissimilarity when $\frac{f}{n} \geq \frac{1}{2+B^2}$.

**Concluding remark.** A critical element to the above inference on the unboundedness of $\varepsilon$ is the condition that $A_1 \leq 0$, shown in (21). Recall that

$$A_1 := \frac{f(n-2f)}{(n-f)^2}\left(\left(1 - \frac{n-2f}{f}B^2\right)\frac{1}{K^2} - 2(1+B^2)\frac{\alpha}{K} + \alpha^2\left(1 - \frac{f}{n-2f}B^2\right)\right) .$$

The right-hand side in the above is negative for any large enough value of $K$ as soon as $1 - \frac{f}{n-2f}B^2 \leq 0$ or, equivalently, $\frac{f}{n} \geq \frac{1}{2+B^2}$. However, in the case when $\frac{f}{n} < \frac{1}{2+B^2}$, $K$ cannot be arbitrarily large if we were to ensure $A_1 \leq 0$, i.e., $K$ must be bounded from above. This constraint on $K$ yields a non-trivial lower bound on $\varepsilon$. We formalize this intuition in the following.

### B.2.2 Lower Bound on $\varepsilon$

In this section, we prove that $\varepsilon \geq \frac{1}{8\mu} \cdot \frac{f}{n-(2+B^2)f} G^2$. Let $0 < f < n/2$, and $G, B \geq 0$. Owing to the arguments presented in Section B.2.1, the assertion holds true when $\frac{f}{n} \geq \frac{1}{2+B^2}$. In the following, we assume that $\frac{f}{n} < \frac{1}{2+B^2}$, or equivalently $B^2 < \frac{n-2f}{f}$.

Let $\mathcal{A}$ be an $(f, \varepsilon)$-resilient algorithm. We consider a distributed learning setting where the workers' loss functions are given by (11), (12) and (13) with parameters $\alpha$, $K$ and $z$ set as follows.

$$\alpha = \mu \left( 1 + B^2 \right), \quad \text{and} \quad K = \frac{1}{\mu \left( 1 - \frac{f}{n-2f} B^2 \right)}. \tag{27}$$

We let $z$ be an arbitrary point in $\mathbb{R}^d$ such that

$$\|z\|^2 = \frac{(n-f)^2}{f(n-2f)} \cdot \frac{G^2}{\alpha(1+B^2)} \cdot K.$$

Note that, as $\alpha = \mu \left( 1 + B^2 \right)$, we have

$$K = \frac{1}{\mu \left( 1 - \frac{f}{n-2f} B^2 \right)} = \frac{1 + B^2}{1 - \frac{f}{n-2f} B^2} \cdot \frac{1}{\alpha}. \tag{28}$$

Therefore,

$$\|z\|^2 = \frac{(n-f)^2}{f(n-2f)} \cdot \frac{1 - \frac{f}{n-2f} B^2}{(1+B^2)^2} \cdot G^2 K^2. \tag{29}$$

From (28), we also obtain that

$$\frac{n-2f}{n-f} \frac{1}{K} + \frac{f}{n-f} \alpha = \frac{n-2f}{n-f} \frac{\left( 1 - \frac{f}{n-2f} B^2 \right)}{1+B^2} \alpha + \frac{f}{n-f} \alpha = \frac{n-2f-fB^2+f+fB^2}{(n-f)(1+B^2)} \alpha$$

$$= \frac{\alpha}{1+B^2}. \tag{30}$$

**Proof outline.** We consider two scenarios, each corresponding to two different identities of honest workers: $S_1 = \{1, \ldots, n-f\}$ and $S_2 = \{f+1, \ldots, n\}$. For each of these, we prove that the corresponding local and global loss functions satisfy the assumptions of the theorem, mainly by invoking Lemma 2. Finally, by invoking Lemma 1, we show that $(f, \varepsilon)$-resilience implies the stated lower bound on $\varepsilon$.

**First scenario.** Consider the set of honest workers to be $S_1 = \{1, \ldots, n-f\}$. From (11) and (12) we obtain that

$$\mathcal{L}_{S_1}(\theta) = \frac{f}{n-f} \frac{\alpha}{2} \|\theta - z\|^2 + \frac{n-2f}{n-f} \frac{1}{2K} \|\theta\|^2. \tag{31}$$

Therefore,

$$\nabla \mathcal{L}_{S_1}(\theta) = \frac{f}{n-f} \alpha (\theta - z) + \frac{n-2f}{n-f} \frac{1}{K} \theta, \quad \text{and} \quad \nabla^2 \mathcal{L}_{S_1}(\theta) = \left( \frac{f}{n-f} \alpha + \frac{n-2f}{n-f} \frac{1}{K} \right) I_d,$$

where $I_d$ denotes the identity matrix of size $d$. The above, in conjunction with (30), implies that $\mathcal{L}_{S_1}$ is $\left( \frac{\alpha}{1+B^2} \right)$-smooth $\left( \frac{\alpha}{1+B^2} \right)$-strongly convex. As $\frac{\alpha}{1+B^2} = \mu$ (see (27)), we deduce that $\mathcal{L}_{S_1}$ is $\mu$-smooth $\mu$-strong convexity. Recall that $\mu \leq L$, therefore $\mathcal{L}_{S_1}$ is also $L$-smooth. Next, by invoking Lemma 2, we show that the local losses for the honest workers in this scenario also satisfy $(G, B)$-dissimilarity.

We start by analyzing below the terms $A_1$, $A_2$ and $A_3$ introduced in Lemma 2 in this scenario.

*Term $A_1$.* Recall from (14) in Lemma 2 that

$$A_1 = \frac{f(n-2f)}{(n-f)^2} \left( \left( 1 - \frac{n-2f}{f} B^2 \right) \frac{1}{K^2} - 2(1+B^2)\frac{\alpha}{K} + \alpha^2 \left( 1 - \frac{f}{n-2f} B^2 \right) \right).$$

Let $A_1' := \frac{(n-f)^2}{f(n-2f)} A_1$. Substituting in the above, from (28), $K = \frac{1+B^2}{1 - \frac{f}{n-2f}B^2} \cdot \frac{1}{\alpha}$, we obtain that

$$A_1' = \left( \left( 1 - \frac{n-2f}{f}B^2 \right) \left( \frac{1 - \frac{f}{n-2f}B^2}{1+B^2} \right)^2 \alpha^2 - 2(1+B^2)\frac{1 - \frac{f}{n-2f}B^2}{1+B^2}\alpha^2 + \alpha^2 \left( 1 - \frac{f}{n-2f}B^2 \right) \right)$$

$$= \alpha^2 \left( 1 - \frac{f}{n-2f}B^2 \right) \left( \frac{\left( 1 - \frac{n-2f}{f}B^2 \right)\left( 1 - \frac{f}{n-2f}B^2 \right)}{(1+B^2)^2} - 1 \right)$$

$$= \alpha^2 \frac{\left( 1 - \frac{f}{n-2f}B^2 \right)}{(1+B^2)^2} \left( \left( 1 - \frac{n-2f}{f}B^2 \right)\left( 1 - \frac{f}{n-2f}B^2 \right) - (1+B^2)^2 \right)$$

$$= \alpha^2 \frac{\left( 1 - \frac{f}{n-2f}B^2 \right)}{(1+B^2)^2} \left( \left( 1 - \frac{n-2f}{f}B^2 - \frac{f}{n-2f}B^2 + B^4 \right) - (1 + 2B^2 + B^4) \right)$$

$$= -\alpha^2 \frac{\left( 1 - \frac{f}{n-2f}B^2 \right)}{(1+B^2)^2} B^2 \left( 2 + \frac{n-2f}{f} + \frac{f}{n-2f} \right) = -\alpha^2 \frac{\left( 1 - \frac{f}{n-2f}B^2 \right) B^2}{(1+B^2)^2} \frac{(n-f)^2}{f(n-2f)} .$$

Recall that $A_1' = \frac{(n-f)^2}{f(n-2f)} A_1$. Therefore, from the above we obtain that

$$A_1 = -\frac{\left( 1 - \frac{f}{n-2f}B^2 \right)}{(1+B^2)^2} B^2 \alpha^2 . \tag{32}$$

*Term $A_2$*. From (14) in Lemma 2, and (30), we obtain that

$$A_2 = \frac{f(n-2f)\alpha}{(n-f)((n-2f)\frac{1}{K} + f\alpha)} \left( \frac{1}{K} - \alpha \right) \frac{1}{K} z = \frac{f(n-2f)}{(n-f)^2} (1+B^2) \left( \frac{1}{K} - \alpha \right) \frac{1}{K} z.$$

From (28), we obtain that $\alpha - \frac{1}{K} = \alpha - \frac{1 - \frac{f}{n-2f}B^2}{1+B^2}\alpha = \frac{n-f}{n-2f} \frac{B^2}{1+B^2}\alpha$. Using this above we obtain that

$$\|A_2\|^2 = \left( \frac{f(n-2f)}{(n-f)^2} \right)^2 (1+B^2)^2 \left( \frac{n-f}{n-2f} \right)^2 \frac{B^4}{(1+B^2)^2} \alpha^2 \frac{\|z\|^2}{K^2} = \left( \frac{f}{n-f} \right)^2 B^4 \alpha^2 \frac{\|z\|^2}{K^2}. \tag{33}$$

*Term $A_3$*. From (14) in Lemma 2, and (30), we obtain that

$$A_3 = \frac{f(n-2f)\alpha^2}{((n-2f)\frac{1}{K} + f\alpha)^2} \frac{\|z\|^2}{K^2} = \frac{f(n-2f)}{(n-f)^2} (1+B^2)^2 \frac{\|z\|^2}{K^2} . \tag{34}$$

Substituting in the above, from (29), $\|z\|^2 = \frac{(n-f)^2}{f(n-2f)} \frac{\left( 1 - \frac{f}{n-2f}B^2 \right)}{(1+B^2)^2} G^2 K^2$ we obtain that

$$A_3 = \left( 1 - \frac{f}{n-2f}B^2 \right) G^2 . \tag{35}$$

*Invoking Lemma 2*. Using the results obtained above we show below that the conditions stated in (15) of Lemma (2) are satisfied, i.e., $A_1 \leq 0$, $A_3 \leq G^2$ and $\|A_2\|^2 \leq A_1(A_3 - G^2)$. Hence, proving that the local loss functions for the honest workers in this particular scenario satisfy $(G, B)$-gradient dissimilarity.

Since we assumed that $B^2 < \frac{n-2f}{f}$, (32) implies that $A_1 \leq 0$. Similarly, (35) implies that $A_3 \leq G^2$. Substituting, from (32) and (34), respectively, $A_1 = -\frac{\left( 1 - \frac{f}{n-2f}B^2 \right)}{(1+B^2)^2} B^2 \alpha^2$ and $A_3 = \frac{f(n-2f)}{(n-f)^2} (1+B^2)^2 \frac{\|z\|^2}{K^2}$, we obtain that

$$A_1(A_3 - G^2) = \frac{\left( 1 - \frac{f}{n-2f}B^2 \right)}{(1+B^2)^2} B^2 \alpha^2 \left( G^2 - \frac{f(n-2f)}{(n-f)^2} (1+B^2)^2 \frac{\|z\|^2}{K^2} \right) .$$

Substituting in the above, from (29), i.e. $\|z\|^2 = \frac{(n-f)^2}{f(n-2f)} \frac{\left(1 - \frac{f}{n-2f} B^2\right)}{(1+B^2)^2} G^2 K^2$, we obtain that

$$A_1(A_3 - G^2) = \frac{\left(1 - \frac{f}{n-2f} B^2\right)}{(1+B^2)^2} B^2 \alpha^2 \left(G^2 - \left(1 - \frac{f}{n-2f} B^2\right) G^2\right)$$

$$= \frac{\left(1 - \frac{f}{n-2f} B^2\right)}{(1+B^2)^2} \left(\frac{f}{n-2f}\right) G^2 B^4 \alpha^2 = \frac{(n-f)^2}{f(n-2f)} \frac{\left(1 - \frac{f}{n-2f} B^2\right)}{(1+B^2)^2} G^2 \left(\frac{f}{n-f}\right)^2 B^4 \alpha^2 \ .$$

Recall that $\frac{(n-f)^2}{f(n-2f)} \frac{\left(1 - \frac{f}{n-2f} B^2\right)}{(1+B^2)^2} G^2 = \frac{\|z\|^2}{K^2}$. Using this above, and then comparing the resulting equation with (33), we obtain that

$$A_1(A_3 - G^2) = \left(\frac{f}{n-f}\right)^2 B^4 \alpha^2 \frac{\|z\|^2}{K^2} = \|A_2\|^2 \ .$$

This concludes the analysis for the first scenario. We have shown that the conditions on smoothness, strong convexity and $(G, B)$-gradient dissimilarity hold in this scenario.

**Second scenario.** Consider the set of honest workers to be $S_2 = \{f+1, \ldots, n\}$. Identical to (31) with $z = 0$, from (12) and (13), we obtain that

$$\mathcal{L}_{S_2}(\theta) = \frac{f}{n-f} \frac{\alpha}{2} \|\theta\|^2 + \frac{n-2f}{n-f} \frac{1}{2K} \|\theta\|^2 \ .$$

Similar to the analysis in the first scenario, we deduce that $\mathcal{L}_{S_2}$ is $L$-smooth and $\mu$-strongly convex. Moreover, Lemma 3, in conjunction with the deduction in (32) that $A_1 \geq 0$ (recall that $B^2 < \frac{n-2f}{f}$), implies that the loss functions for the honest workers in this particular scenario satisfy $(0, B)$-gradient dissimilarity, thereby also satisfying $(G, B)$-gradient dissimilarity.

This concludes the analysis for the second scenario. We have shown that the conditions on smoothness, strong convexity and $(G, B)$-gradient dissimilarity hold in this scenario.

**Final step: lower bound on $\varepsilon$ in terms of $G$ and $B$.** We have established in the above that the conditions of smoothness, strong convexity and $(G, B)$-gradient dissimilarity are satisfied in both the scenarios. Therefore, by assumptions, algorithm $\mathcal{A}$ must guarantee $(f, \varepsilon)$-resilience in either scenario. Specifically, the output of $\mathcal{A}$, denoted by $\hat{\theta}$ must satisfy the following:

$$\max\left\{\mathcal{L}_{S_1}(\hat{\theta}) - \mathcal{L}_{*, S_1}, \mathcal{L}_{S_2}(\hat{\theta}) - \mathcal{L}_{*, S_2}\right\} \leq \varepsilon.$$

Due to Lemma 1, the above holds only if

$$\varepsilon \geq \frac{\left(\frac{f}{n-f}\right)^2 \alpha^2}{8\left(\frac{n-2f}{n-f} \frac{1}{K} + \frac{f}{n-f} \alpha\right)} \|z\|^2 \ . \tag{36}$$

Recall, from (30), that $\frac{n-2f}{n-f} \frac{1}{K} + \frac{f}{n-f} \alpha = \frac{\alpha}{1+B^2}$. Using this in (36), we obtain that

$$\varepsilon \geq \frac{\left(\frac{f}{n-f}\right)^2 \alpha^2}{8\left(\frac{\alpha}{1+B^2}\right)} \|z\|^2 = \frac{1}{8}\left(\frac{f}{n-f}\right)^2 \alpha(1 + B^2) \|z\|^2$$

Substituting, from (27), $\alpha = \mu(1 + B^2)$ in the above implies that

$$\varepsilon \geq \frac{\mu}{8}\left(\frac{f}{n-f}\right)^2 (1 + B^2)^2 \|z\|^2 \ .$$

Substituting, from (29), $\|z\|^2 = \frac{(n-f)^2}{f(n-2f)} \frac{\left(1 - \frac{f}{n-2f} B^2\right)}{(1+B^2)^2} G^2 K^2$ in the above implies that

$$\varepsilon \geq \frac{\mu}{8}\left(\frac{f}{n-f}\right)^2 (1 + B^2)^2 \frac{(n-f)^2}{f(n-2f)} \frac{\left(1 - \frac{f}{n-2f} B^2\right)}{(1+B^2)^2} G^2 K^2$$

$$= \frac{\mu}{8}\left(\frac{f}{n-2f}\right)\left(1 - \frac{f}{n-2f} B^2\right) G^2 K^2 \ .$$

Substituting, from (27), i.e. $K = \frac{1}{\mu\left(1 - \frac{f}{n-2f}B^2\right)}$, in the above implies that

$$\varepsilon \geq \frac{\mu}{8}\left(\frac{f}{n-2f}\right)\left(1 - \frac{f}{n-2f}B^2\right)\frac{G^2}{\mu^2\left(1 - \frac{f}{n-2f}B^2\right)^2} = \frac{1}{8\mu}\cdot\frac{\frac{f}{n-2f}G^2}{1 - \frac{f}{n-2f}B^2}$$

The above completes the proof.

### B.3  Proof of Lemma 1

Let us recall the lemma below.

**Lemma 1.** *Consider the setting where the local loss functions are given by (11), (12), and (13). In this particular case, the following holds for all $\theta \in \mathbb{R}^d$:*

$$\max\left\{\mathcal{L}_{S_1}(\theta) - \mathcal{L}_{*,S_1},\ \mathcal{L}_{S_2}(\theta) - \mathcal{L}_{*,S_2}\right\} \geq \frac{\left(\frac{f}{n-f}\right)^2\alpha^2}{8\left(\frac{n-2f}{n-f}\frac{1}{K} + \frac{f}{n-f}\alpha\right)}\|z\|^2\ .$$

*Proof.* We prove the lemma by contradiction. Suppose there exists a parameter vector $\hat{\theta}$ such that

$$\max\left\{\mathcal{L}_{S_1}(\hat{\theta}) - \mathcal{L}_{*,S_1},\ \mathcal{L}_{S_2}(\hat{\theta}) - \mathcal{L}_{*,S_2}\right\} < \frac{\left(\frac{f}{n-f}\right)^2\alpha^2}{8\left(\frac{n-2f}{n-f}\frac{1}{K} + \frac{f}{n-f}\alpha\right)}\|z\|^2 =: \delta\ . \tag{37}$$

From (11), (12) and (13), we obtain that

$$\mathcal{L}_{S_1}(\theta) := \frac{1}{n-f}\left(f\frac{\alpha}{2}\|\theta - z\|^2 + (n - 2f)\frac{1}{2K}\|\theta\|^2\right)\ ,\ \text{and}$$

$$\mathcal{L}_{S_2}(\theta) :==\frac{1}{n-f}\left(f\frac{\alpha}{2}\|\theta\|^2 + (n - 2f)\frac{1}{2K}\|\theta\|^2\right)\ .$$

Therefore, we have[4]

$$\mathcal{L}_{S_1}(\hat{\theta}) - \mathcal{L}_{*,S_1} = \frac{1}{2}\left(\frac{n-2f}{n-f}\frac{1}{K} + \frac{f}{n-f}\alpha\right)\left\|\hat{\theta} - \frac{f\alpha}{(n-2f)\frac{1}{K} + f\alpha}z\right\|^2\ ,\ \text{and}$$

$$\mathcal{L}_{S_2}(\hat{\theta}) - \mathcal{L}_{*,S_2} = \frac{1}{2}\left(\frac{n-2f}{n-f}\frac{1}{K} + \frac{f}{n-f}\alpha\right)\left\|\hat{\theta}\right\|^2\ .$$

Substituting from the above in (37), we obtain that

$$\delta > \frac{1}{2}\left(\frac{n-2f}{n-f}\frac{1}{K} + \frac{f}{n-f}\alpha\right)\max\left\{\left\|\hat{\theta} - \frac{f\alpha}{(n-2f)\frac{1}{K} + f\alpha}z\right\|^2,\left\|\hat{\theta}\right\|^2\right\}.$$

As for any real values $a, b$, we have $\max\{a,\ b\} \geq \frac{1}{2}(a + b)$, from above we obtain that

$$\delta > \frac{1}{4}\left(\frac{n-2f}{n-f}\frac{1}{K} + \frac{f}{n-f}\alpha\right)\left(\left\|\hat{\theta} - \frac{f\alpha}{(n-2f)\frac{1}{K} + f\alpha}z\right\|^2 + \left\|\hat{\theta}\right\|^2\right). \tag{38}$$

By triangle and Jensen's inequalities, we have

$$\left\|\frac{f\alpha}{(n-2f)\frac{1}{K} + f\alpha}z\right\|^2 = \left\|\frac{f\alpha}{(n-2f)\frac{1}{K} + f\alpha}z - \hat{\theta} + \hat{\theta}\right\|^2 \leq 2\left\|\hat{\theta} - \frac{f\alpha}{(n-2f)\frac{1}{K} + f\alpha}z\right\|^2 + 2\left\|\hat{\theta}\right\|^2\ .$$

---

[4]For arbitrary positive real values $a$ and $b$, and an arbitrary $z \in \mathbb{R}^d$, consider a loss function $\mathcal{L}(\theta) := \frac{a}{2}\|\theta - z\|^2 + \frac{b}{2}\|\theta\|^2$. The minimum point $\theta_*$ of $\mathcal{L}(\theta)$ is given by $\theta_* = \frac{a}{a+b}z$, and for any $\theta$, $\mathcal{L}(\theta) - \mathcal{L}(\theta_*) = \frac{1}{2}(a + b)\|\theta - \theta_*\|^2$.

Substituting from the above in (38), we obtain that

$$\delta > \frac{1}{8} \left( \frac{n-2f}{n-f}\frac{1}{K} + \frac{f}{n-f}\alpha \right) \left\| \frac{f\alpha}{(n-2f)\frac{1}{K} + f\alpha} z \right\|^2$$

$$= \frac{1}{8} \left( \frac{n-2f}{n-f}\frac{1}{K} + \frac{f}{n-f}\alpha \right) \left\| \frac{\frac{f}{n-f}\alpha}{\frac{n-2f}{n-f}\frac{1}{K} + \frac{f}{n-f}\alpha} z \right\|^2 = \frac{\left(\frac{f}{n-f}\right)^2 \alpha^2}{8\left(\frac{n-2f}{n-f}\frac{1}{K} + \frac{f}{n-f}\alpha\right)} \|z\|^2 = \delta .$$

The contradiction above proves the lemma. $\qquad\square$

## B.4   Proof of Lemma 2

Let us recall the lemma below.

**Lemma 2.** *Consider the setting where the local loss functions are given by* (11)*,* (12)*, and* (13)*. Denote*

$$A_1 := \frac{f(n-2f)}{(n-f)^2} \left( \left(1 - \frac{n-2f}{f}B^2\right) \frac{1}{K^2} - 2(1+B^2)\frac{\alpha}{K} + \left(1 - \frac{f}{n-2f}B^2\right)\alpha^2 \right) ,$$

$$A_2 := \frac{f(n-2f)\alpha}{(n-f)((n-2f)\frac{1}{K} + f\alpha)} \left( \frac{1}{K} - \alpha \right) \frac{1}{K} z , \text{ and } A_3 := \frac{f(n-2f)\alpha^2}{\left((n-2f)\frac{1}{K} + f\alpha\right)^2} \frac{\|z\|^2}{K^2} .$$

$$(14)$$

*Suppose that* $S_1 = \{1, \dots, n-f\}$ *denotes the set of honest workers. Then, the honest workers satisfy* $(G, B)$*-gradient dissimilarity if and only if*

$$A_1 \le 0, \qquad A_3 \le G^2, \quad and \quad \|A_2\|^2 \le A_1(A_3 - G^2). \tag{15}$$

*Proof.* Let $\theta \in \mathbb{R}^d$. As $\frac{1}{|S_1|}\sum_{i\in S_1}\|\nabla\mathcal{L}_i(\theta) - \nabla\mathcal{L}_{S_1}(\theta)\|^2 = \frac{1}{|S_1|}\sum_{i\in S_1}\|\nabla\mathcal{L}_i(\theta)\|^2 - \|\nabla\mathcal{L}_{S_1}(\theta)\|^2$, we obtain that

$$\frac{1}{|S_1|}\sum_{i\in S_1}\|\nabla\mathcal{L}_i(\theta) - \nabla\mathcal{L}_{S_1}(\theta)\|^2 - \left(G^2 + B^2\|\nabla\mathcal{L}_{S_1}(\theta)\|^2\right)$$

$$= \frac{1}{|S_1|}\sum_{i\in S_1}\|\nabla\mathcal{L}_i(\theta)\|^2 - (1 + B^2)\|\nabla\mathcal{L}_{S_1}(\theta)\|^2 - G^2. \tag{39}$$

We analyze the right-hand side of the above equality. As $S_1 = \{1, \dots, n-f\}$, from (11) and (12), we obtain that

$$\frac{1}{|S_1|}\sum_{i\in S_1}\|\nabla\mathcal{L}_i(\theta)\|^2 = \frac{n-2f}{n-f}\|\nabla\mathcal{L}_{\mathrm{II}}(\theta)\|^2 + \frac{f}{n-f}\|\nabla\mathcal{L}_{\mathrm{I}}(\theta)\|^2$$

$$= \frac{n-2f}{n-f}\frac{1}{K^2}\|\theta\|^2 + \frac{f}{n-f}\alpha^2\|\theta - z\|^2. \tag{40}$$

Similarly, we have

$$\nabla\mathcal{L}_{S_1}(\theta) = \frac{1}{|S_1|}\sum_{i\in S_1}\nabla\mathcal{L}_i(\theta) = \frac{n-2f}{n-f}\frac{1}{K}\theta + \frac{f}{n-f}\alpha\,(\theta - z)$$

$$= \frac{(n-2f)\frac{1}{K} + f\alpha}{n-f}\theta - \frac{f}{n-f}\alpha\,z = \underbrace{\frac{(n-2f)\frac{1}{K} + f\alpha}{n-f}}_{A_0} \left( \theta - \underbrace{\frac{f\alpha}{(n-2f)\frac{1}{K} + f\alpha}z}_{\theta_*} \right) .$$

Denoting in the above

$$A_0 := \frac{(n-2f)\frac{1}{K} + f\alpha}{n-f} \tag{41}$$

and

$$\theta_* := \frac{f\alpha}{(n-2f)\frac{1}{K}+f\alpha} z \,, \tag{42}$$

we have

$$\nabla\mathcal{L}_{S_1}(\theta) = A_0\left(\theta - \theta_*\right).$$

The above implies that $\theta_*$ is the minimizer of the convex function $\mathcal{L}_{S_1}$, and

$$\left\|\nabla\mathcal{L}_{S_1}(\theta)\right\|^2 = A_0^2\left\|\theta - \theta_*\right\|^2. \tag{43}$$

Substituting from (40) and (43) in (39) implies that

$$\frac{1}{|S_1|}\sum_{i\in S_1}\left\|\nabla\mathcal{L}_i(\theta)-\nabla\mathcal{L}_{S_1}(\theta)\right\|^2 - G^2 - B^2\left\|\nabla\mathcal{L}_{S_1}(\theta)\right\|^2$$

$$= \frac{n-2f}{n-f}\frac{1}{K^2}\left\|\theta\right\|^2 + \frac{f}{n-f}\alpha^2\left\|\theta - z\right\|^2 - (1+B^2)A_0^2\left\|\theta - \theta_*\right\|^2 - G^2.$$

Now, we operate the change of variables $X = \theta - \theta_*$, and rewrite the above as

$$\frac{1}{|S_1|}\sum_{i\in S_1}\left\|\nabla\mathcal{L}_i(\theta)-\nabla\mathcal{L}_{S_1}(\theta)\right\|^2 - G^2 - B^2\left\|\nabla\mathcal{L}_{S_1}(\theta)\right\|^2$$

$$= \frac{n-2f}{n-f}\frac{1}{K^2}\left\|X+\theta_*\right\|^2 + \frac{f}{n-f}\alpha^2\left\|X+\theta_* - z\right\|^2 - (1+B^2)A_0^2\left\|X\right\|^2 - G^2$$

$$= \frac{n-2f}{n-f}\frac{1}{K^2}\left(\left\|X\right\|^2 + \left\|\theta_*\right\|^2 + 2\left\langle X, \theta_*\right\rangle\right) + \frac{f}{n-f}\alpha^2\left(\left\|X\right\|^2 + \left\|\theta_* - z\right\|^2 + 2\left\langle X, \theta_* - z\right\rangle\right)$$

$$- (1+B^2)A_0^2\left\|X\right\|^2 - G^2$$

$$= \left(\underbrace{\frac{(n-2f)\frac{1}{K^2}+f\alpha^2}{n-f} - (1+B^2)A_0^2}_{A_1}\right)\left\|X\right\|^2 + 2\left\langle X, \underbrace{\frac{n-2f}{n-f}\frac{1}{K^2}\theta_* + \frac{f}{n-f}\alpha^2(\theta_* - z)}_{A_2}\right\rangle$$

$$+ \underbrace{\frac{n-2f}{n-f}\frac{1}{K^2}\left\|\theta_*\right\|^2 + \frac{f}{n-f}\alpha^2\left\|\theta_* - z\right\|^2}_{A_3} - G^2. \tag{44}$$

Next, we show that $A_1$, $A_2$ and $A_3$ as defined in (14) can be equivalently written as follows.

$$A_1 = \frac{(n-2f)\frac{1}{K^2}+f\alpha^2}{n-f} - (1+B^2)A_0^2, \quad A_2 = \frac{n-2f}{n-f}\frac{1}{K^2}\theta_* + \frac{f}{n-f}\alpha^2(\theta_* - z) \quad \text{and}$$

$$A_3 = \frac{n-2f}{n-f}\frac{1}{K^2}\left\|\theta_*\right\|^2 + \frac{f}{n-f}\alpha^2\left\|\theta_* - z\right\|^2 \,.$$

**Term $A_1$:** Substituting from (41), i.e. $A_0 = \frac{(n-2f)\frac{1}{K}+f\alpha}{n-f}$, we obtain that

$$\frac{(n-2f)\frac{1}{K^2}+f\alpha^2}{n-f} - (1+B^2)A_0^2 = \frac{(n-2f)\frac{1}{K^2}+f\alpha^2}{n-f} - (1+B^2)\left(\frac{(n-2f)\frac{1}{K}+f\alpha}{n-f}\right)^2$$

$$= \frac{(n-2f)\frac{1}{K^2}+f\alpha^2}{n-f} - (1+B^2)\left(\left(\frac{n-2f}{n-f}\right)^2\frac{1}{K^2} + \left(\frac{f}{n-f}\right)^2\alpha^2 + \frac{2f(n-2f)}{(n-f)^2}\frac{\alpha}{K}\right)$$

$$= \left(\frac{n-2f}{n-f}\right)^2\left(\frac{n-f}{n-2f} - (1+B^2)\right)\frac{1}{K^2} - 2(1+B^2)\frac{f(n-2f)}{(n-f)^2}\frac{\alpha}{K}$$

$$+ \left(\frac{f}{n-f}\right)^2\alpha^2\left(\frac{n-f}{f} - (1+B^2)\right)$$

$$= \left(\frac{n-2f}{n-f}\right)^2\left(\frac{f}{n-2f} - B^2\right)\frac{1}{K^2} - 2(1+B^2)\frac{f(n-2f)}{(n-f)^2}\frac{\alpha}{K} + \left(\frac{f}{n-f}\right)^2\alpha^2\left(\frac{n-2f}{f} - B^2\right)$$

$$= \frac{f(n-2f)}{(n-f)^2}\left(\left(1 - \frac{n-2f}{f}B^2\right)\frac{1}{K^2} - 2(1+B^2)\frac{\alpha}{K} + \left(1 - \frac{f}{n-2f}B^2\right)\alpha^2\right) \,.$$

Comparing the above with $A_1$ defined in (14) implies that

$$\frac{(n-2f)\frac{1}{K^2}+f\alpha^2}{n-f} - (1+B^2)A_0^2 = A_1 \,. \tag{45}$$

**Term $A_2$:** Substituting from (42), i.e. $\theta_* = \frac{f\alpha}{(n-2f)\frac{1}{K}+f\alpha}z$, we obtain that

$$\frac{n-2f}{n-f}\frac{1}{K^2}\theta_* + \frac{f}{n-f}\alpha^2(\theta_*-z)$$

$$= \frac{n-2f}{n-f}\frac{1}{K^2}\frac{f\alpha}{(n-2f)\frac{1}{K}+f\alpha}z + \frac{f}{n-f}\alpha^2\left(\frac{f\alpha}{(n-2f)\frac{1}{K}+f\alpha}z-z\right)$$

$$= \frac{n-2f}{n-f}\frac{1}{K^2}\frac{f\alpha}{(n-2f)\frac{1}{K}+f\alpha}z - \frac{f}{n-f}\alpha^2\frac{(n-2f)\frac{1}{K}}{(n-2f)\frac{1}{K}+f\alpha}z$$

$$= \frac{f(n-2f)\alpha}{(n-f)((n-2f)\frac{1}{K}+f\alpha)}\left(\frac{1}{K}-\alpha\right)\frac{1}{K}z \,.$$

Comparing the above with $A_2$ defined in (14) implies that

$$\frac{n-2f}{n-f}\frac{1}{K^2}\theta_* + \frac{f}{n-f}\alpha^2(\theta_*-z) = A_2 \,. \tag{46}$$

**Term $A_3$:** Similarly, substituting $\theta_* = \frac{f\alpha}{(n-2f)\frac{1}{K}+f\alpha}z$, we obtain that

$$\frac{n-2f}{n-f}\frac{1}{K^2}\|\theta_*\|^2 + \frac{f}{n-f}\alpha^2\|\theta_*-z\|^2$$

$$= \frac{n-2f}{n-f}\frac{1}{K^2}\left(\frac{f\alpha}{(n-2f)\frac{1}{K}+f\alpha}\right)^2\|z\|^2 + \frac{f}{n-f}\alpha^2\left(\frac{f\alpha}{(n-2f)\frac{1}{K}+f\alpha}-1\right)^2\|z\|^2$$

$$= \frac{n-2f}{n-f}\frac{1}{K^2}\left(\frac{f\alpha}{(n-2f)\frac{1}{K}+f\alpha}\right)^2\|z\|^2 + \frac{f}{n-f}\alpha^2\left(\frac{(n-2f)\frac{1}{K}}{(n-2f)\frac{1}{K}+f\alpha}\right)^2\|z\|^2$$

$$= \frac{f(n-2f)\alpha^2}{\left((n-2f)\frac{1}{K}+f\alpha\right)^2}\frac{\|z\|^2}{K^2} \,.$$

Comparing the above with $A_3$ defined in (14) implies that

$$\frac{n-2f}{n-f}\frac{1}{K^2}\|\theta_*\|^2 + \frac{f}{n-f}\alpha^2\|\theta_*-z\|^2 = A_3 \,. \tag{47}$$

Substituting from (45), (46) and (47) in (44), we obtain that

$$\frac{1}{|S_1|}\sum_{i\in S_1}\|\nabla\mathcal{L}_i(\theta)-\nabla\mathcal{L}_{S_1}(\theta)\|^2 - G^2 - B^2\|\nabla\mathcal{L}_{S_1}(\theta)\|^2 = A_1\|X\|^2 + 2\langle X, A_2\rangle + A_3 - G^2.$$

Therefore, by Assumption 1, the honest workers represented by set $S_1$ satisfy $(G,B)$-gradient dissimilarity if and only if the right-hand side of the above equations is less than or equal to 0, i.e.,

$$A_1\|X\|^2 + 2\langle X, A_2\rangle + A_3 - G^2 \le 0 \,, \quad \text{for all } X = \theta - \theta_* \in \mathbb{R}^d \,. \tag{48}$$

To show the above, we consider an auxiliary second-order "polynomial" in $x \in \mathbb{R}^d$, $P(x) :=  a\|x\|^2 + 2\langle x, b\rangle + c$ where $b \in \mathbb{R}^d$ and $a,c \in \mathbb{R}$. We show below that $P(x) \le 0$ for all $x \in \mathbb{R}^d$ if and only if $a,c \le 0$ and $\Delta := \|b\|^2 - ac \le 0$.

*Case 1.* Let $a \ne 0$. In this particular case, we have $P(x) = a\left(\|x+\frac{1}{a}\cdot b\|^2 - \frac{\Delta}{a^2}\right)$. Therefore, $P(x) \le 0$ for all $x$ if and only if $a < 0$ and $c, \Delta \le 0$.

*Case 2.* Let $a = 0$. In this particular case, $P(x) = 2\langle x, b\rangle + c$. Therefore, $P(x) \le 0$ for all $x$ if and only if $c \le 0$ and $\Delta = \|b\|^2 = 0$.

Hence, (48) is equivalent to

$$A_1 \le 0 \,, \qquad A_3 - G^2 \le 0 \,, \quad \text{and} \quad \|A_2\|^2 - A_1(A_3 - G^2) \le 0 \,.$$

This completes the proof. $\qquad\square$

# C  Proofs of Theorem 2 and Corollary 1: Convergence Results

## C.1  Proof of Theorem 2

**Theorem 2.** *Let $0 \leq f < n/2$. Assume that the global loss $\mathcal{L}_{\mathcal{H}}$ is $L$-smooth and that the honest local losses satisfy $(G, B)$-gradient dissimilarity (Assumption 1). Consider Algorithm 1 with learning rate $\gamma = \frac{1}{L}$. If the aggregation $F$ is $(f, \kappa)$-robust with $\kappa B^2 < 1$, then the following holds for all $T \geq 1$.*

1. *In the general case where $\mathcal{L}_{\mathcal{H}}$ may be non-convex, we have*

$$\frac{1}{T} \sum_{t=0}^{T-1} \|\nabla \mathcal{L}_{\mathcal{H}}(\theta_t)\|^2 \leq \frac{\kappa G^2}{1 - \kappa B^2} + \frac{2L \left(\mathcal{L}_{\mathcal{H}}(\theta_0) - \mathcal{L}_{*,\mathcal{H}}\right)}{(1 - \kappa B^2)T}.$$

2. *In the case where $\mathcal{L}_{\mathcal{H}}$ is $\mu$-PL, we have*

$$\mathcal{L}_{\mathcal{H}}(\theta_T) - \mathcal{L}_{*,\mathcal{H}} \leq \frac{\kappa G^2}{2\mu \left(1 - \kappa B^2\right)} + e^{-\frac{\mu}{L}\left(1 - \kappa B^2\right)T} \left(\mathcal{L}_{\mathcal{H}}(\theta_0) - \mathcal{L}_{*,\mathcal{H}}\right).$$

### C.1.1  Non-convex Case

*Proof.* As $\mathcal{L}_{\mathcal{H}}$ is assumed $L$-smooth, for all $\theta$, $\theta' \in \mathbb{R}^d$, we have (see Definition 2)

$$\mathcal{L}_{\mathcal{H}}(\theta') - \mathcal{L}_{\mathcal{H}}(\theta) \leq \langle \nabla \mathcal{L}_{\mathcal{H}}(\theta), \, \theta' - \theta \rangle + \frac{L}{2} \|\theta' - \theta\|^2 .$$

Let $t \in \{0, \ldots, T-1\}$. From Algorithm 1, recall that $\theta_{t+1} = \theta_t - \gamma R_t$. Hence, substituting in the above $\theta = \theta_t$ and $\theta' = \theta_{t+1}$, we have

$$\mathcal{L}_{\mathcal{H}}(\theta_{t+1}) - \mathcal{L}_{\mathcal{H}}(\theta_t) \leq -\gamma \langle \nabla \mathcal{L}_{\mathcal{H}}(\theta_t), \, R_t \rangle + \frac{1}{2}\gamma^2 L \|R_t\|^2 . \tag{49}$$

As $\langle a, \, b \rangle = \frac{1}{2} \left( \|a\|^2 + \|b\|^2 - \|a - b\|^2 \right)$ for any $a, b \in \mathbb{R}^d$, we also have

$$\langle \nabla \mathcal{L}_{\mathcal{H}}(\theta_t), \, R_t \rangle = \frac{1}{2} \left( \|\nabla \mathcal{L}_{\mathcal{H}}(\theta_t)\|^2 + \|R_t\|^2 - \|\nabla \mathcal{L}_{\mathcal{H}}(\theta_t) - R_t\|^2 \right).$$

Substituting the above in (49) we obtain that

$$\mathcal{L}_{\mathcal{H}}(\theta_{t+1}) - \mathcal{L}_{\mathcal{H}}(\theta_t) \leq -\frac{\gamma}{2} \left( \|\nabla \mathcal{L}_{\mathcal{H}}(\theta_t)\|^2 + \|R_t\|^2 - \|\nabla \mathcal{L}_{\mathcal{H}}(\theta_t) - R_t\|^2 \right) + \frac{1}{2}\gamma^2 L \|R_t\|^2$$

$$= -\frac{\gamma}{2} \|\nabla \mathcal{L}_{\mathcal{H}}(\theta_t)\|^2 - \frac{\gamma}{2}(1 - \gamma L) \|R_t\|^2 + \frac{\gamma}{2} \|\nabla \mathcal{L}_{\mathcal{H}}(\theta_t) - R_t\|^2 .$$

Substituting $\gamma = \frac{1}{L}$ in the above we obtain that

$$\mathcal{L}_{\mathcal{H}}(\theta_{t+1}) - \mathcal{L}_{\mathcal{H}}(\theta_t) \leq -\frac{1}{2L} \|\nabla \mathcal{L}_{\mathcal{H}}(\theta_t)\|^2 + \frac{1}{2L} \|R_t - \nabla \mathcal{L}_{\mathcal{H}}(\theta_t)\|^2 . \tag{50}$$

As we assume the aggregation $F$ to satisfy $(f, \kappa)$-robustness, by Definition 4, we also have that

$$\|R_t - \nabla \mathcal{L}_{\mathcal{H}}(\theta_t)\|^2 = \left\| F\left(g_t^{(1)}, \ldots, g_t^{(n)}\right) - \frac{1}{|\mathcal{H}|} \sum_{i \in \mathcal{H}} g_t^{(i)} \right\|^2 \leq \frac{\kappa}{|\mathcal{H}|} \sum_{i \in \mathcal{H}} \left\| g_t^{(i)} - \frac{1}{|\mathcal{H}|} \sum_{i \in \mathcal{H}} g_t^{(i)} \right\|^2$$

$$= \frac{\kappa}{|\mathcal{H}|} \sum_{i \in \mathcal{H}} \|\nabla \mathcal{L}_i(\theta_t) - \nabla \mathcal{L}_{\mathcal{H}}(\theta_t)\|^2 . \tag{51}$$

Besides, Assumption 1 implies that for all $\theta \in \mathbb{R}^d$ we have

$$\frac{1}{|\mathcal{H}|} \sum_{i \in \mathcal{H}} \|\nabla \mathcal{L}_i(\theta) - \nabla \mathcal{L}_{\mathcal{H}}(\theta)\|^2 \leq G^2 + B^2 \|\nabla \mathcal{L}_{\mathcal{H}}(\theta)\|^2 .$$

Using the above in (51) yields

$$\|R_t - \nabla \mathcal{L}_{\mathcal{H}}(\theta_t)\|^2 \leq \kappa G^2 + \kappa B^2 \|\nabla \mathcal{L}_{\mathcal{H}}(\theta_t)\|^2 .$$

Substituting the above in (50) yields

$$\mathcal{L}_{\mathcal{H}}(\theta_{t+1}) - \mathcal{L}_{\mathcal{H}}(\theta_t) \leq -\frac{1}{2L} \|\nabla \mathcal{L}_{\mathcal{H}}(\theta_t)\|^2 + \frac{1}{2L} \left( \kappa G^2 + \kappa B^2 \|\nabla \mathcal{L}_{\mathcal{H}}(\theta_t)\|^2 \right). \tag{52}$$

Multiplying both sides in (52) by $2L$ and rearranging the terms, we get

$$\left( 1 - \kappa B^2 \right) \|\nabla \mathcal{L}_{\mathcal{H}}(\theta_t)\|^2 \leq \kappa G^2 + 2L \left( \mathcal{L}_{\mathcal{H}}(\theta_t) - \mathcal{L}_{\mathcal{H}}(\theta_{t+1}) \right). \tag{53}$$

Recall that $t$ in the above was arbitrary in $\{0, \dots, T-1\}$. Averaging over all $t \in \{0, \dots, T-1\}$ yields

$$\left( 1 - \kappa B^2 \right) \frac{1}{T} \sum_{t=0}^{T-1} \|\nabla \mathcal{L}_{\mathcal{H}}(\theta_t)\|^2 \leq \kappa G^2 + \frac{2L}{T} \sum_{t=0}^{T-1} \left( \mathcal{L}_{\mathcal{H}}(\theta_t) - \mathcal{L}_{\mathcal{H}}(\theta_{t+1}) \right) = \kappa G^2 + \frac{2L}{T} \left( \mathcal{L}_{\mathcal{H}}(\theta_0) - \mathcal{L}_{\mathcal{H}}(\theta_T) \right)$$

$$\leq \kappa G^2 + \frac{2L}{T} \left( \mathcal{L}_{\mathcal{H}}(\theta_0) - \mathcal{L}_{*,\mathcal{H}} \right).$$

Finally, since we assume that $1 - \kappa B^2 > 0$, dividing both sides in the above by $1 - \kappa B^2$ yields

$$\frac{1}{T} \sum_{t=0}^{T-1} \|\nabla \mathcal{L}_{\mathcal{H}}(\theta_t)\|^2 \leq \frac{\kappa G^2}{1 - \kappa B^2} + \frac{2L \left( \mathcal{L}_{\mathcal{H}}(\theta_0) - \mathcal{L}_{*,\mathcal{H}} \right)}{(1 - \kappa B^2)T}.$$

The above concludes the proof for the non-convex case. $\qquad \square$

### C.1.2 Strongly Convex Case

*Proof.* Assume now that $\mathcal{L}_{\mathcal{H}}$ is $\mu$-PL. Following the proof of the non-convex case up until (53) yields

$$\left( 1 - \kappa B^2 \right) \|\nabla \mathcal{L}_{\mathcal{H}}(\theta_t)\|^2 \leq \kappa G^2 + 2L \left( \mathcal{L}_{\mathcal{H}}(\theta_t) - \mathcal{L}_{\mathcal{H}}(\theta_{t+1}) \right)$$

$$= \kappa G^2 + 2L \left( \mathcal{L}_{\mathcal{H}}(\theta_t) - \mathcal{L}_{*,\mathcal{H}} + \mathcal{L}_{*,\mathcal{H}} - \mathcal{L}_{\mathcal{H}}(\theta_{t+1}) \right).$$

Rearranging terms, we get

$$2L \left( \mathcal{L}_{\mathcal{H}}(\theta_{t+1}) - \mathcal{L}_{*,\mathcal{H}} \right) \leq \kappa G^2 - \left( 1 - \kappa B^2 \right) \|\nabla \mathcal{L}_{\mathcal{H}}(\theta_t)\|^2 + 2L \left( \mathcal{L}_{\mathcal{H}}(\theta_t) - \mathcal{L}_{*,\mathcal{H}} \right).$$

Since $\mathcal{L}_{\mathcal{H}}$ is $\mu$-PL, as per Definition 3, we obtain

$$2L \left( \mathcal{L}_{\mathcal{H}}(\theta_{t+1}) - \mathcal{L}_{*,\mathcal{H}} \right) \leq \kappa G^2 - 2\mu \left( 1 - \kappa B^2 \right) \left( \mathcal{L}_{\mathcal{H}}(\theta_t) - \mathcal{L}_{*,\mathcal{H}} \right) + 2L \left( \mathcal{L}_{\mathcal{H}}(\theta_t) - \mathcal{L}_{*,\mathcal{H}} \right)$$

$$= \kappa G^2 + \left( 2L - 2\mu \left( 1 - \kappa B^2 \right) \right) \left( \mathcal{L}_{\mathcal{H}}(\theta_t) - \mathcal{L}_{*,\mathcal{H}} \right).$$

Dividing both sides by $2L$, we get

$$\mathcal{L}_{\mathcal{H}}(\theta_{t+1}) - \mathcal{L}_{*,\mathcal{H}} \leq \frac{\kappa G^2}{2L} + \left( 1 - \frac{\mu}{L} \left( 1 - \kappa B^2 \right) \right) \left( \mathcal{L}_{\mathcal{H}}(\theta_t) - \mathcal{L}_{*,\mathcal{H}} \right). \tag{54}$$

Recall that $t$ is arbitrary in $\{0, \dots, T-1\}$. Then, applying (54) recursively (on the right hand side) yields

$$\mathcal{L}_{\mathcal{H}}(\theta_{t+1}) - \mathcal{L}_{*,\mathcal{H}} \leq \frac{\kappa G^2}{2L} \sum_{k=0}^{t} \left( 1 - \frac{\mu}{L} \left( 1 - \kappa B^2 \right) \right)^k$$

$$+ \left( 1 - \frac{\mu}{L} \left( 1 - \kappa B^2 \right) \right)^{t+1} \left( \mathcal{L}_{\mathcal{H}}(\theta_0) - \mathcal{L}_{*,\mathcal{H}} \right)$$

$$\leq \frac{\kappa G^2}{2L} \frac{1}{1 - \left( 1 - \frac{\mu}{L} \left( 1 - \kappa B^2 \right) \right)} + \left( 1 - \frac{\mu}{L} \left( 1 - \kappa B^2 \right) \right)^{t+1} \left( \mathcal{L}_{\mathcal{H}}(\theta_0) - \mathcal{L}_{*,\mathcal{H}} \right)$$

$$= \frac{\kappa G^2}{2\mu \left( 1 - \kappa B^2 \right)} + \left( 1 - \frac{\mu}{L} \left( 1 - \kappa B^2 \right) \right)^{t+1} \left( \mathcal{L}_{\mathcal{H}}(\theta_0) - \mathcal{L}_{*,\mathcal{H}} \right).$$

Using the fact that $(1+x)^n \leq e^{nx}$ for all $x \in \mathbb{R}$ and substituting $t = T-1$ yields

$$\mathcal{L}_{\mathcal{H}}(\theta_T) - \mathcal{L}_{*,\mathcal{H}} \leq \frac{\kappa G^2}{2\mu \left( 1 - \kappa B^2 \right)} + e^{-\frac{\mu}{L} \left( 1 - \kappa B^2 \right) T} \left( \mathcal{L}_{\mathcal{H}}(\theta_0) - \mathcal{L}_{*,\mathcal{H}} \right).$$

This concludes the proof. $\qquad \square$

## C.2  Proof of Corollary 1

**Corollary 1.** *Assume that the global loss $\mathcal{L}_{\mathcal{H}}$ is $\mu$-PL and $L$-smooth, and that for each $i \in \mathcal{H}$ local loss $\mathcal{L}_i$ is convex and $L_i$-smooth. Denote $L_{\max} := \max_{i \in \mathcal{H}} L_i$ and assume that $\kappa(\frac{3L_{\max}}{\mu} - 1) \leq 1$. Consider Algorithm 1 with learning rate $\gamma = \frac{1}{L}$. If $F$ is $(f, \kappa)$-robust, then for all $T \geq 1$, we have*

$$\mathcal{L}_{\mathcal{H}}(\theta_T) - \mathcal{L}_{*,\mathcal{H}} \leq \frac{3\kappa}{\mu} \frac{1}{|\mathcal{H}|} \sum_{i \in \mathcal{H}} \|\nabla \mathcal{L}_i(\theta_*)\|^2 + e^{-\frac{\mu}{3L}T} \left(\mathcal{L}_{\mathcal{H}}(\theta_0) - \mathcal{L}_{*,\mathcal{H}}\right).$$

*Proof.* Invoking Proposition 1, we know that the loss functions satisfy $(G, B)$-gradient dissimilarity with

$$G^2 = \frac{2}{|\mathcal{H}|} \sum_{i \in \mathcal{H}} \|\nabla \mathcal{L}_i(\theta_*)\|^2, \qquad B^2 = 2\frac{L_{\max}}{\mu} - 1.$$

Moreover, since we assume that $\kappa(3\frac{L_{\max}}{\mu} - 1) \leq 1$, we have

$$1 - \kappa B^2 = 1 - \kappa(2\frac{L_{\max}}{\mu} - 1) = 1 - \frac{2}{3}\kappa(3\frac{L_{\max}}{\mu} - 1) + \frac{\kappa}{3} \geq 1 - \frac{2}{3} + \frac{\kappa}{3} \geq \frac{1}{3}.$$

Therefore, since the global loss $\mathcal{L}_{\mathcal{H}}$ is $\mu$-PL and $L$-smooth, we can apply Theorem 2 to obtain

$$\begin{aligned}
\mathcal{L}_{\mathcal{H}}(\theta_T) - \mathcal{L}_{*,\mathcal{H}} &\leq \frac{\kappa G^2}{2\mu(1 - \kappa B^2)} + e^{-\frac{\mu}{L}(1 - \kappa B^2)T} \left(\mathcal{L}_{\mathcal{H}}(\theta_0) - \mathcal{L}_{*,\mathcal{H}}\right) \\
&\leq \frac{3}{2\mu}\kappa G^2 + e^{-\frac{\mu}{3L}T} \left(\mathcal{L}_{\mathcal{H}}(\theta_0) - \mathcal{L}_*\right) \\
&= \frac{3\kappa}{\mu} \frac{1}{|\mathcal{H}|} \sum_{i \in \mathcal{H}} \|\nabla \mathcal{L}_i(\theta_*)\|^2 + e^{-\frac{\mu}{3L}T} \left(\mathcal{L}_{\mathcal{H}}(\theta_0) - \mathcal{L}_{*,\mathcal{H}}\right).
\end{aligned}$$

This concludes the proof.

$\square$

# D  Experimental Setups

In this section, we present the full experimental setups of the experiments in Figures 1,2,3, and 4. All our experiments were conducted on the following hardware: Macbook Pro, Apple M1 chip, 8-core CPU and 2 NVIDIA A10-24GB GPUs. Our code is available online through this link.

## D.1  Figure 1: brittleness of $G$-Gradient Dissimilarity

This first experiment aims to show the gap between existing theory and practice. While in theory $G$-gradient dissimilarity does not cover the least square regression problem, we show that it is indeed possible to converge using the Robust D-GD Algorithm with the presence of 3 Byzantine workers out of 10 total workers. All the hyperparameters of this experiments are listed in Table 1.

| | |
|---|---|
| **Number of Byzantine workers** | $f = 3$ |
| **Number of honest workers** | $n - f = 7$ |
| **Dataset** | $n - f$ datapoints in *mg* LIBSVM [10] selected uniformly without replacement |
| **Data heterogeneity** | Each honest worker holds one distinct point |
| **Model** | Linear regression |
| **Algorithm** | Robust D-GD |
| **Number of steps** | $T = 40000$ |
| **Learning rate** | $\gamma = 0.001$ |
| **Loss function** | Regularized Mean Least Square error |
| **$\ell_2$-regularization term** | $\lambda = 1/\sqrt{m(n-f)} = 1/\sqrt{7}$ |
| **Aggregation rule** | NNM [3] coupled with CW Trimmed Mean [34] |
| **Byzantine attacks** | *sign flipping* [2], *fall of empires* [33], *a little is enough* [5] and *mimic* [20] |

Table 1: Setup of Figure 1's experiment

## D.2 Figure 2: empirical breakdown point

The second experiment we conduct tends to highlight the empirical breakdown point observed in practice when the fraction of Byzantines becomes too high. Indeed, while existing theory suggests that this breakdown point occurs for a fraction of $1/2$ Byzantine workers, we show empirically that it occurs even before having $1/4$ Byzantines. We present all the hyperparameters used for this experiment in Table 2.

| | |
|---|---|
| **Number of Byzantine workers** | from $f = 1$ to $f = 9$ |
| **Number of honest workers** | $n - f = 10$ |
| **Dataset** | 10% of MNIST selected uniformly without replacement |
| **Data heterogeneity** | Each worker dataset holds data from a distinct class |
| **Model** | Logistic regression |
| **Algorithm** | Robust D-GD |
| **Number of steps** | $T = 500$ |
| **Learning rate** | $\gamma = \begin{cases} 0.05 & \text{if} \quad 0 \leq T < 350 \\ 0.01 & \text{if} \quad 350 \leq T < 420 \\ 0.002 & \text{if} \quad 420 \leq T < 480 \\ 0.0004 & \text{if} \quad 480 \leq T < 500 \end{cases}$ |
| **Loss function** | Negative Log Likelihood (NLL) |
| $\ell_2$**-regularization term** | $10^{-4}$ |
| **Aggregation rule** | NNM [3] & CW Trimmed Mean [34], Krum [7], CW Median [34] or geometric median [30] |
| **Byzantine attacks** | *sign flipping* [2] |

Table 2: Setup of Figure 2's experiment

### D.3 Figure 3: comparing theoretical upper bounds

In the third experiment, we compare two errors bounds, i.e., $f/n-(2+B^2)f \cdot G^2$ and $f/n-2f \cdot \widehat{G}^2$, guaranteed for robust D-GD under $(G, B)$-gradient dissimilarity and $\widehat{G}$-gradient dissimilarity, respectively, on the MNIST Dataset. We use a logistic regression model and the Negative Log Likelihood (NLL) loss function as the local loss function for each honest worker.

Before explaining how we compute $\widehat{G}$ and $(G, B)$, let us recall that the local loss functions of honest workers are said to satisfy $\widehat{G}$-gradient dissimilarity if for all $\theta \in \mathbb{R}^d$, we have

$$\frac{1}{|\mathcal{H}|} \sum_{i \in \mathcal{H}} \|\nabla \mathcal{L}_i(\theta) - \nabla \mathcal{L}_\mathcal{H}(\theta)\|^2 \leq \widehat{G}^2 .$$

Similarly, the local loss functions of honest workers are said to satisfy $(G, B)$-*gradient dissimilarity* if, for all $\theta \in \mathbb{R}^d$, we have

$$\frac{1}{|\mathcal{H}|} \sum_{i \in \mathcal{H}} \|\nabla \mathcal{L}_i(\theta) - \nabla \mathcal{L}_\mathcal{H}(\theta)\|^2 \leq G^2 + B^2 \|\nabla \mathcal{L}_\mathcal{H}(\theta)\|^2 .$$

Evidently, $\widehat{G}$ and $(G, B)$ are difficult to compute since one has to explore the entire space to get tight values. In this paper, we present a first heuristic to compute approximate values of $\widehat{G}$ and $(G, B)$: We first compute $\theta_\star$ by running D-GD without Byzantine workers, then we choose a point $\theta_0$, that is arbitrarily far from $\theta_\star$. Here we choose $\theta_0 = (1e4, \ldots, 1e4) \in \mathbb{R}^d$ with $d = 784$ for the MNIST dataset. Given $\theta_\star$ and $\theta_0$ we construct 101 points $\theta_0, \ldots, \theta_{100}$ between $\theta_\star$ and $\theta_0$ such that

$$\theta_t = \frac{t}{100} \times \theta_\star + \left(1 - \frac{t}{100}\right) \times \theta_0 , \quad \forall t \in \{0, 1, 2, \ldots, 100\} ,$$

and then compute $\frac{1}{|\mathcal{H}|} \sum_{i \in \mathcal{H}} \|\nabla \mathcal{L}_i(\theta_t) - \nabla \mathcal{L}_\mathcal{H}(\theta_t)\|^2$ and $\|\nabla \mathcal{L}_\mathcal{H}(\theta_t)\|^2$ for all $t \in \{0, 1, 2, \ldots, 100\}$. We set

$$\widehat{G}^2 = \max_{t \in \{0,1,2,\ldots,100\}} \frac{1}{|\mathcal{H}|} \sum_{i \in \mathcal{H}} \|\nabla \mathcal{L}_i(\theta_t) - \nabla \mathcal{L}_\mathcal{H}(\theta_t)\|^2 .$$

Also, to compute the couple $(G, B)$, we first consider 100 values of $B'^2 \in \left[0, \frac{n-2f}{f}\right]$, then compute

$$G_{B'}^2 = \max_{t \in \{0,1,2,\ldots,100\}} \frac{1}{|\mathcal{H}|} \sum_{i \in \mathcal{H}} \|\nabla \mathcal{L}_i(\theta_t) - \nabla \mathcal{L}_\mathcal{H}(\theta_t)\|^2 - B'^2 \|\nabla \mathcal{L}_\mathcal{H}(\theta_t)\|^2 .$$

and finally set

$$(G, B) = \underset{(G_{B'}, B')}{\arg \min} \frac{f}{n - (2 + B'^2)f} \cdot G_{B'}^2 .$$

In Figure 3, we show the different values of $f/n-(2+B^2)f \cdot G^2$ and $f/n-2f \cdot \widehat{G}^2$ for $f = 1$ to $f = 9$ where there is always $n - f = 10$ honest workers.

## D.4 Figure 4: matching empirical performances

In the last experiments of the paper, we compare the empirical error gap (left-hand side of Corollary 1) with the upper bound (right-hand side of Corollary 1).

Let us first recall Corollary 1:

**Corollary 1.** *Assume that the global loss $\mathcal{L}_{\mathcal{H}}$ is $\mu$-PL and $L$-smooth, and that for each $i \in \mathcal{H}$ local loss $\mathcal{L}_i$ is convex and $L_i$-smooth. Denote $L_{\max} := \max_{i \in \mathcal{H}} L_i$ and assume that $\kappa(\frac{3L_{\max}}{\mu} - 1) \leq 1$. Consider Algorithm 1 with learning rate $\gamma = \frac{1}{L}$. If $F$ is $(f, \kappa)$-robust, then for all $T \geq 1$, we have*

$$\mathcal{L}_{\mathcal{H}}(\theta_T) - \mathcal{L}_{*,\mathcal{H}} \leq \frac{3\kappa}{\mu} \frac{1}{|\mathcal{H}|} \sum_{i \in \mathcal{H}} \|\nabla \mathcal{L}_i(\theta_*)\|^2 + e^{-\frac{\mu}{3L}T} \left( \mathcal{L}_{\mathcal{H}}(\theta_0) - \mathcal{L}_{*,\mathcal{H}} \right).$$

We first give all the hyperparameters used for the learning in Table 3 and then explain how we computed $\mu$, $L$, $\kappa$ and $\theta_\star$.

| | |
|---|---|
| **Number of Byzantine workers** | from $f = 1$ to $f = 19$ |
| **Number of honest workers** | $n - f = 20$ |
| **Dataset** | $n - f$ datapoints in *mg* LIBSVM [10] selected uniformly without replacement |
| **Data heterogeneity** | Each honest worker holds $m = 1$ distinct point |
| **Model** | Linear regression |
| **Algorithm** | Robust D-GD |
| **Number of steps** | $T = 40000$ |
| **Learning rate** | $\gamma = 0.0001$ |
| **Loss function** | Regularized Mean Least Square error |
| **$\ell_2$-regularization term** | $\lambda = 1/\sqrt{m(n-f)} = 1/\sqrt{7}$ |
| **Aggregation rule** | NNM [3] coupled with CW Trimmed Mean [34] |
| **Byzantine attacks** | *sign flipping* [2], *fall of empires* [33], *a little is enough* [5] and *mimic* [20] |

Table 3: Setup of Figure 4's experiment

**Computation of $\mu$ and $L$.** Let $\mathbf{X}$ be the matrix that contains all the $n - f$ vector data points and $\lambda$ the $\ell_2$-regularization term, we have

$$L = \text{eig}_{\max}\left( \frac{1}{n-f} \mathbf{X}^\top \mathbf{X} + \lambda \mathbf{I} \right) \quad , \quad \mu = \text{eig}_{\min}\left( \frac{1}{n-f} \mathbf{X}^\top \mathbf{X} + \lambda \mathbf{I} \right) \tag{55}$$

where for any matrix $\mathbf{A} \in \mathbb{R}^{d \times d}$, $\text{eig}_{\max}(\mathbf{A})$ and $\text{eig}_{\max}(\mathbf{A})$ refer to the maximum and minimum eigenvalue of $\mathbf{A}$ respectively.

**Computation of $\kappa$.** We recall that by definition, an aggregation rule $F \colon \mathbb{R}^{d \times n} \to \mathbb{R}^d$ is said to be $(f, \kappa)$-*robust* if for any vectors $x_1, \ldots, x_n \in \mathbb{R}^d$, and any set $S \subseteq [n]$ of size $n - f$, the output $\hat{x} := F(x_1, \ldots, x_n)$ satisfies the following:

$$\|\hat{x} - \overline{x}_S\|^2 \leq \kappa \cdot \frac{1}{|S|} \sum_{i \in S} \|x_i - \overline{x}_S\|^2,$$

where $\overline{x}_S := \frac{1}{|S|} \sum_{i \in S} x_i$.

Hence, as done in [3], we estimate $\kappa$ empirically as follows: We first compute for every step $t \in \{0, \ldots, T-1\}$ and every attack $a \in \{\text{SF}, \text{FOE}, \text{ALIE}, \text{Mimic}\}$ the value of $\kappa_{t,a}$ such that

$$\kappa_{t,a} = \frac{\left\| R_{t,a} - \overline{g}_{t,a} \right\|^2}{\frac{1}{|S|} \sum_{i \in S} \left\| g_{t,a}^{(i)} - \overline{g}_{t,a} \right\|^2}$$

where $\overline{g}_{t,a} := \frac{1}{|\mathcal{H}|} \sum_{i \in \mathcal{H}} g_{t,a}^{(i)}$ and $R_{t,a}$ and $g_{t,a}^{(i)}$ refer respectively to $R_t$ and $g_t^{(i)}$ when the attack $a$ is used by the Byzantine workers. Then, we compute the empirical $\kappa$ following:

$$\kappa = \max_{\substack{t \in \{0,\ldots,T-1\} \\ a \in \{\text{SF},\text{FOE},\text{ALIE},\text{Mimic}\}}} \kappa_{t,a}.$$

**Computation of $\theta_\star$.** We compute $\theta_\star$ using the closed form of the solution of a Mean Least Square regression problem:

$$\theta_\star = \left( \mathbf{X}^\top \mathbf{X} + (n - f)\, \lambda \mathbf{I} \right)^{-1} \mathbf{X}^\top \mathbf{y},$$

where $\mathbf{X} \in \mathbb{R}^{(n-f) \times d}$ is the matrix that contains the data points and $\mathbf{y} \in \mathbb{R}^{n-f}$ the associated vector that contains the labels.

