# OpenReview forum: "Robust Distributed Learning: Tight Error Bounds and Breakdown Point under Data Heterogeneity"
_NeurIPS.cc/2023/Conference — NeurIPS 2023 spotlight_

### Official Review · Reviewer_YywB · 2023-07-05

**Soundness:** 3 good
**Presentation:** 3 good
**Contribution:** 2 fair
**Rating:** 7
**Confidence:** 3

**Summary:**

The paper studies distributed learning in the presense of Byzantine workers (that is, nodes that output adversarial results) and data heterogeneity.

A previous notion of robustness for distributed learning, namely G-gradient dissimilarity, is not sufficient to capture data heterogeneity. In fact, this holds even in fairly simple tasks such as least-squares regression (i.e., for any G>0).

Motivated by this observation, the paper defines a new notion of robustness for distributed learning, (G,B)-Gradient Dissimilarity. This allows for some additional error term in the average gradient difference between workers, proportionally to B.

The proposed theory is used in a least squares regression task to justify observed robustness of stochastic gradient descent under small sets of dishonest workers.

**Strengths:**

The paper presents an interesting notion of robustness in distributed learning. It is shown that this notion captures distributed gradient computation when a small number of workers can output arbitrary results. The theory is compelling and seems to justify observed behavior in a realistic setting.

A theoretical criterion is given for algorithms to be robust against (G,B)-gradient dissimilarity. This is an interesting theoretical contribution.


**Weaknesses:**

The precise formula used in the definition of (G,B)-Gradient Dissimilarity makes sense because the analysis is done for a variant of stochastic gradient descent. However, it is unclear whether this notion is also relevant to other distributed learning tasks.

The precise setting regarding honest and Byzantine workers is confusing (see Questions section of this review).


**Questions:**

What is the (G, B)-gradient dissimilarity in the experiments?

Observation 1 does not require any Byzantine workers, right? Perhaps this should be emphasized.

In Proposition 1, H is the set of honest workers. How do the honest workers know who is honest? Earlier in the paper it is mentioned that trying to compute any statistic over the whole set of workers is generally impossible in the presence of Byzantine workers. Does this mean that the task in Proposition 1 is impossible to perform in reality because H must be known to all honest workers? If so, this renders the whole model not very realistic.


**Limitations:**

The authors have adequately addressed the limitations of this work.

---

> ### Author Rebuttal · Authors · 2023-08-08
>
> Thank you for the encouraging comments. Please find below answers to your questions.
>
> >What is the (G, B)-gradient dissimilarity in the experiments?
>
> Intuitively, $(G, B)$-gradient dissimilarity measures data heterogeneity using the gradient of the empirical loss function. For a given distributed learning problem, there may be multiple couples $(G,B)$ for which the gradient dissimilarity inequality holds (naive example: if it holds for $(G,B)$, then it also holds for $(G,2B)$). To show the tightest empirical upper bounds (Figures 3 and 4), based on our result in Theorem 2, we search through several possible values of $(G,B)$ using a heuristic method explained in Appendix D.
>
> >Observation 1 does not require any Byzantine workers, right? Perhaps this should be emphasized.
>
> Yes, we agree. Thank you for pointing this out. We will emphasize it in the paper.
>
> >In Proposition 1, H is the set of honest workers. How do the honest workers know who is honest? Earlier in the paper it is mentioned that trying to compute any statistic over the whole set of workers is generally impossible in the presence of Byzantine workers. Does this mean that the task in Proposition 1 is impossible to perform in reality because H must be known to all honest workers? If so, this renders the whole model not very realistic.
>
> Thank you for bringing up this point. Honest workers and the server do not know the set $\mathcal{H}$. This is in fact one of the main theoretical arguments in the proof of the lower bound of Theorem 1. Of course, in Theorem 2, we analyze the convergence of Robust D-GD for the loss function over the set of honest workers $\mathcal{H}$.
>
> >The precise formula used in the definition of (G,B)-Gradient Dissimilarity makes sense because the analysis is done for a variant of stochastic gradient descent. However, it is unclear whether this notion is also relevant to other distributed learning tasks.
>
> This is a good remark, as one could think of the applicability of our findings to higher order optimization algorithms for example. In this sense, extending $(G,B)$-Gradient Dissimilarity to incorporate higher-order information is an interesting research direction. However, as long as the loss is differentiable we believe that $(G,B)$-Gradient Dissimilarity is the primary heterogeneity model to study, as was done before in the references in lines 143-144. Finally, we emphasize that, under $(G,B)$-Gradient Dissimilarity, the lower bound in Theorem 1 applies to any algorithm, and not just a specific variant of gradient descent. That is, we do not assume any structure of the algorithm for demonstrating Theorem 1.

---

### Official Review · Reviewer_UheS · 2023-07-06

**Soundness:** 3 good
**Presentation:** 3 good
**Contribution:** 3 good
**Rating:** 6
**Confidence:** 2

**Summary:**

This paper considers under the robust distributed learning settings, the conventional learning error bounds are vacuous as they rely on restrictive assumptions as $G$-gradient dissimilarity (equation 2). The paper relaxes the assumption to a more general $(G, B)$-dissimilarity model that relaxes the bounds on gradients norm of honest workers with a constant of $B$. A new breakdown ratio between Byzantine and honest workers $f / n$ that would yield theoretical robustness guarantees is established as $ 1 / (2 + B^2) $. The authors finally show the tightness of $f / n < 1 / (2 + B^2)$ bound via analyzing the state of the art robustness coefficient $ \kappa = f / (n - 2f) $ on a robust D-GD that would also yield $ 1 / (2 + B^2) $ instead of $ 1 / 2 $ breakdown point. In this way, the gap between the previous theory and the empirical observation of the largest amount of Byzantine workers to have robustness guarantee on global honest loss (equation 1) is reduced.

**Strengths:**

The paper borrows the data heterogenity model ($(G, B)$-dissimilarity) from the classical distributed learning settings to the Byzantine distributed learning and build a nice and intuitive theory on the breakdown point on the fraction of Byzantine workers to melt down the robustness guarantee. The paper narratives are clear and easy to follow, and the experiments (Figure 2, 3, 4) are well conducted.

**Weaknesses:**

The novel breakpoint essentially involves a dependency over the constant over honest workers' gradient norm. It would be better to have an ablation study on logistic regression that if such constant $B$ is changed, we would observe similar changes in the breakpoint point. Otherwise, it is a great paper!

**Questions:**

I am confused on the heuristic method to compute $G$ and $B$ in Appendix D.3, line 743. Could you explain such formula?

**Limitations:**

I didn't find any other limitations of this paper.

---

> ### Author Rebuttal · Authors · 2023-08-08
>
> Thank you for the encouraging comments. Please find below answers to your questions. We hope that you would increase the score if you find our answers satisfactory.
>
> >I am confused on the heuristic method to compute and in Appendix D.3, line 743. Could you explain such formula?
>
> First, we recall that a fixed set of loss functions can satisfy the condition of $(G,B)$-gradient dissimilarity for multiple pairs $(G,B)$. For example, if the loss functions satisfy $(G,B)$-gradient dissimilarity, they also satisfy $(G,2B)$-gradient dissimilarity. Thus, our heuristic finds multiple couples $(G,B)$ and chooses that yielding the smallest upper bound (as per Theorem 2). To do so, we consider several possible values of $B$ (whose range is bounded as per the lower bound in Theorem 1). Then, for each possible value of $B$, we find the corresponding tightest $G$ satisfying Assumption 1 (as described in line 742). Finally, in line 743, we find the pair $(G, B)$ minimizing the theoretical upper bound.
>
> >The novel breakpoint essentially involves a dependency over the constant over honest workers' gradient norm. It would be better to have an ablation study on logistic regression that if such constant $B$ is changed, we would observe similar changes in the breakpoint point. Otherwise, it is a great paper!
>
> This is an interesting remark. It is however quite challenging to conduct such an ablation study. This is mainly because we cannot directly control the constant $B$, subject to Assumption 1, even for simple learning problems like least-squares regression. Thus, in our experiments, we first partition data heterogeneously across workers, and then determine $B$ empirically.

---

> > ### Comment · Reviewer_UheS · 2023-08-14
> > **Reply to the rebuttal**
> >
> > Thank you for the clarification. I now understand the estimation approach on $B$ in Appendix D.3.

---

> > > ### Author Response · Authors · 2023-08-14
> > >
> > > Thank you very much for your timely response. We hope all your concerns have been addressed at this point. We will be glad to answer any further question that may lead you to increasing your score.

---

### Official Review · Reviewer_VidP · 2023-07-11

**Soundness:** 3 good
**Presentation:** 3 good
**Contribution:** 3 good
**Rating:** 7
**Confidence:** 3

**Summary:**

The work focuses on the analysis of the limits of Byzantine-robust learning under so-called $(G,B)$-dissimilarity assumption -- a generalization of the standard bounded gradient dissimilarity assumption. The main result of the paper is in establishing the upper bound for the maximal admissible ratio of Byzantine workers: $\delta < \frac{1}{2+B^2}$ that is separated from $\frac{1}{2}$ (and can be very small). The authors also derive the lower bound for the optimization error that matches the known upper bounds (up to numerical factors) and the derived upper bound for the robust version of the distributed gradient descent.

**Strengths:**

1. The authors derive the limits on robustness under $(G,B)$-gradient dissimilarity that are quite interesting. In particular, the ratio of Byzantine workers has to be smaller than $\frac{1}{2+B^2}$ to guarantee the convergence to some neighborhood of the solution. Moreover, the lower bound for the optimization error $\varepsilon$ is established. These bounds are quite pessimistic ($B^2$ is proportional to the condition number in the worst case) and emphasize the non-triviality of Byzantine-robustness in the heterogeneous case. The construction of the worst-case examples is quite simple.

2. The paper is well-written in general, the idea is explained well in the main part of the paper.

3. The proofs look correct to me.

**Weaknesses:**

The results of Theorem 2 are not completely novel. In particular, there exist results under $(G,B)$-dissimilarity assumption, see Theorem V from [18] and Theorems E.1, E.2 from [14]. These results are derived for the different notions of robust aggregator, but up to the replacement of $\kappa$ with $c\delta$ they recover Theorem 2 from this submission. Moreover, the method from [14] with $p=1$ and no compression coincides with Robust D-GD.

The paper should be modified accordingly. Nevertheless, I see Theorem 1 as the main contribution.

**Questions:**

1. In [17,18], another notion of robust aggregation was introduced. Are there any benefits of using a different notion of robustness?

2. Theorem 2, "let $f < n/2$": this is a bit confusing, because $\kappa$ implicitly depends on the ratio of Byzantines.

3. Although the proofs are detailed and look correct to me, it is a bit inconvenient to read them. In particular, I suggest moving the proofs of Lemmas 1 and 2 to Appendix B.1.2.

**Limitations:**

The authors adequately addressed the limitations.

---

> ### Author Rebuttal · Authors · 2023-08-08
>
> Thank you for the encouraging comments. We will implement your suggestions in the final version of the paper. Please find below answers to your questions.
>
> >The results of Theorem 2 are not completely novel. In particular, there exist results under $(G,B)$-dissimilarity assumption, see Theorem V from [18] and Theorems E.1, E.2 from [14]. These results are derived for the different notions of robust aggregator, but up to the replacement of $\kappa$ with $c \delta$ they recover Theorem 2 from this submission. Moreover, the method from [14] with $p=1$ and no compression coincides with Robust D-GD.
>
>  We thank the reviewer for bringing up this point, as we were aware of these results. While we agree on the similarities between the results, there are subtle differences that are important to note. First, unlike the notion of $(f,\kappa)$-robustness (that we use), the so-called $(c,\delta)$-agnostic robustness (used in [14, 18]) is a stochastic notion. Under the latter notion, good parameters $(c,\delta)$ of robust aggregators were only shown when using a randomized method called Bucketing [18]. Consequently, instead of obtaining a deterministic error bound as in Theorem 2, simply replacing $c\delta$ with $\kappa$ in [14, 18] gives a stochastic bound, which is strictly weaker than the result of Theorem 2. Moreover, the corresponding non-vanishing upper bound term for robust D-GD obtained from the analysis in [18] for several robust aggregation rules (e.g., coordinate-wise median) is worse than that we obtain using $(f,\kappa)$-robustness (see also discussion below). We will include this discussion in the paper.
>
> >In [17,18], another notion of robust aggregation was introduced. Are there any benefits of using a different notion of robustness?
>
> The main benefit of $(f,\kappa)$-robustness (the notion of robust aggregation that we use) is that it was shown (originally in [3]) to hold for several existing aggregation rules (e.g. coordinate-wise median, trimmed mean) with tight rates of $\kappa$, unlike $(c, \delta)$-agnostic robustness introduced in [17, 18]. Also, as argued in [3], existing robustness criteria in the literature can be recovered from $(f,\kappa)$-robustness. Importantly, as pointed out above, $(f,\kappa)$-robustness is a deterministic condition, which allows us to obtain a deterministic bound for robust D-GD (unlike in [18]).
>
> >Theorem 2, ``let $f < n/2$": this is a bit confusing, because $\kappa$ implicitly depends on the ratio of Byzantines.
>
> The reason we explicitly assume $f < n/2$ in the beginning of the theorem is that no upper bound can be achieved without this condition, as shown in [25]. We then let $F$ be a $(f,\kappa)$-robust aggregation, which is implicit for ``let $F \colon \mathbb{R}^{d \times n} \to \mathbb{R}$ and $\kappa \geq 0$ be such that $F$ is $(f,\kappa)$-robust". The theorem then holds for any $\kappa$ verifying this condition (exact values for $\kappa$ can be found in [3]).

---

> > ### Comment · Reviewer_VidP · 2023-08-14
> > **Thank you for the clarifications**
> >
> > I thank the authors for the clarifications: all my questions are adequately addressed. Since the authors promised to modify the paper accordingly, I decided to keep my score unchanged.

---

### Official Review · Reviewer_GcGL · 2023-07-26

**Soundness:** 4 excellent
**Presentation:** 4 excellent
**Contribution:** 4 excellent
**Rating:** 8
**Confidence:** 4

**Summary:**

In the present paper the authors study the problem of distributed learning in the presence of adversarial learners (aka Byzantine workers) and hetogeneous data. The paper aims to generalize and improve previous results on homogeneous data, an assumption that is claimed to be very restrictive in practice. For this they introduce a new concept of (G,B) gradient dissimilarity which has the same relation to the more standard G gradient dissimilarity as an affine variance assumptions has to a bounded variance assumption in stochastic optimization.

Under their assumption they first prove that the breakdown point, i.e., the share of Byzantine workers beyond which the optimization breaks down is upper bounded by $\tfrac{1}{2+B^2}$, where previous results only yield the intuitive bound of $\tfrac12$. They also confirm this theoretical observation numerically. Second, they also establish sharp error bounds in the regime below the breakdown point by proving a lower bound as well as an upper bound, attained by robust distributed gradient descent.



**Strengths:**

I find the paper very clear and well written. The theoretical contributions are very interesting and appear to be novel. Furthermore, the experiments illustrate the sharpness of the results. The proofs in the appendix seem to be sound although I have to say that I didn't check all the details in the proof of Theorem 1. I did check the convergence statement of Theorem 2 the proof of which is rather standard.

**Weaknesses:**

I cannot list any significant weaknesses besides the few questions that I have in the next block.

**Questions:**

p.2: Maybe you can make the connection between the G and (G,B) assumptions and bounded / affine variance assumptions in the analysis of SGD.

p.6, last paragraph: can one estimate B here to see whether it is close to $\sqrt{2}$ which would fit to the $\tfrac14$ you observe numerically?

p.7: Definition 4 should be followed by a few examples for $(f, \kappa)$-robust aggregation rules. Please also mention that in the absence of adversaries, one can just average the gradients and gets a $(0,0)$-robust aggregation rule.

p.8: Here you start using nicefrac excessively which, at least for my brain, is very hard to read and process. Maybe you can use or stick to tfrac for inline fractions which you seemed to have used earlier.

p.30: Number of honest workers should just be $n-f$ not $n-f=10$. If I understand this correctly $n=10$ and $f$ ranges from $1$ to $9$.



**Limitations:**

The authors did make the point that probably not all of their results are sharp, e.g., in terms of constants or the slow down factor for linear convergence in the presence of adversaries.
Negative societal impact is not to be expected in my opinion.

---

> ### Author Rebuttal · Authors · 2023-08-08
>
> Thank you for the encouraging comments. We will implement your suggestions in the final version of the paper. Please find below answers to your questions.
>
> > p.2: Maybe you can make the connection between the G and (G,B) assumptions and bounded / affine variance assumptions in the analysis of SGD.
>
> This is a good remark. We will make the connection between these in the final version of the paper.
>
> >p.6, last paragraph: Can one estimate B here to see whether it is close to $\sqrt{2}$ which would fit to the $\frac{1}{4}$ you observe numerically?
>
>  This is an interesting question. We believe it is possible to get the result mentioned, similar to the way we measured $G$ and $B$ empirically in Figures 3 and 4. We will include this result in the final version.
>
> >p.30: Number of honest workers should just be $n-f$ and not $n-f = 10$. If I understand this correctly $n = 10$ and $f$ ranger from $1$ to $9$.
>
> In experiments, we fix the number of honest workers and vary the number of Byzantine workers. In this particular case, the number of honest workers (i.e., $n-f$) is set to $10$, and the number of Byzantine workers $f$ varies from $1$ to $9$.

---

> > ### Comment · Reviewer_GcGL · 2023-08-10
> > **Reply to rebuttal**
> >
> > Thanks for the clarifications. It would be great if you could also add the examples after Definition 4 but besides that I'm happy with the submission and will maintain my score.

---

> > > ### Author Response · Authors · 2023-08-11
> > >
> > > Thanks for your response! We will add examples after Definition 4 in the revision of the paper.

---

> > > > ### Comment · Reviewer_GcGL · 2023-08-11
> > > >
> > > > Great thanks!

---

### Decision · Program_Chairs · 2023-09-21

**Decision:**

Accept (spotlight)

**Comment:**

The paper advances our understanding of Byzantine resilient Distributed learning. Under a natural assumption of inhomogeneity that extends the standard one, it shows new breakdown point for Byzantine resilient learning, and establishes new lower and upper bounds for problems.

This is a clear accept paper.